# Dissecting the High Esterase/Lipase Activity and Probiotic Traits in *Lactiplantibacillus plantarum* B22: A Genome-Guided Functional Characterization

**DOI:** 10.3390/foods14132354

**Published:** 2025-07-02

**Authors:** Yunmei Chai, Zhenzhu Li, Wentao Zheng, Xue Yang, Jinze He, Shaomei Hu, Jindou Shi, Yufang Li, Guangqiang Wei, Aixiang Huang

**Affiliations:** Department of Food Science, College of Food Science and Technology, Yunnan Agricultural University, Kunming 650201, China; yunmeichai@163.com (Y.C.); ly03020909@163.com (Z.L.); wentaozheng9986@163.com (W.Z.); 18428000763@163.com (X.Y.); hejinze1123@163.com (J.H.); 18687254936@163.com (S.H.); sh488709976@163.com (J.S.); yufangfangli@126.com (Y.L.); guangqiangwei@126.com (G.W.)

**Keywords:** *Lactiplantibacillus plantarum* B22, esterase, lipase, genomic analysis, probiotic properties, safety

## Abstract

*Lactiplantibacillus plantarum* B22 exhibits a high esterase/lipase activity, but the genomic and probiotic potential remains unclear. We employed an integrated approach combining whole-genome sequencing, molecular docking studies, and phenotypic assays to dissect the genomic and functional basis underlying the high lipolytic activity and probiotic traits of *L.plantarum* B22. This strain exhibited a robust lipase activity (3.45 ± 0.13 U/mL), with whole-genome analysis revealing that the complete genome of this strain spans 2,027,325 bp, encoding 2005 genes with a guanine-cytosine (GC) content of 35.06%. Notably, 13 esterase/lipase genes were identified, 4 of which (gene3060, gene3059, gene2553, gene0798) harbor conserved catalytic triads (Ser-His-Gly/Ala), essential for lipase function. Molecular docking studies confirmed strong binding affinity to tributyrin (ΔG ≤ –5.52 kcal/mol) and elucidated the interaction mechanisms, involving hydrogen bonding and hydrophobic interactions between the esterase/lipase enzymes and tributyrin. Phenotypic and genomic analyses further demonstrated that *L. plantarum* B22 possesses excellent tolerance to simulated human gastrointestinal tract conditions, along with potent antioxidant and antimicrobial activities, highlighting its strong probiotic potential. Genomic annotation also identified 68 genes associated with lipid metabolism and an intact fatty acid synthesis pathway. Importantly, the analysis of phenotypes and genes involved in virulence factors, and the production of harmful metabolites suggests that *L. plantarum* B22 is safe. Collectively, this study offers novel insights into the genome-guided functional characterization of *L. plantarum* B22, providing a robust foundation for its development as a functional probiotic strain.

## 1. Introduction

Carboxylesterases are hydrolytic enzymes that catalyze the cleavage of ester bonds in carboxylic acid esters. Based on their substrate specificity and catalytic mechanisms, they are classified into two major groups: lipases (EC 3.1.1.3) and esterases (EC 3.1.1.1) [1]. Lipase, also known as glycerol ester hydrolase or triacylglycerol acyl hydrolase, primarily catalyzes the hydrolysis of triglycerides into glycerol and fatty acids, thereby enhancing the nutritional value and flavor of the product [2]. Esterases preferentially act on short-chain carboxylic esters (C2–C6), yielding carboxylic acids and alcohols through hydrolysis. Carboxylate ester hydrolase has been widely used in the food industry [3]. Compared to chemical catalysts, esterase/lipase functions under milder conditions and exhibits good biodegradability [4]. These properties make lipase an attractive candidate for various applications. Esterases and lipases are mainly sourced from animals, plants, and microorganisms [5]. Microbial lipases offer numerous advantages, including greater diversity, faster proliferation, and ease of cultivation [6]. They also exhibit a wider operational range concerning pH, temperature, and substrate specificity [6]. These advantages render microbial lipases suitable for industrial-scale production. Microbial lipases are primarily derived from fungi and bacteria, such as *Aspergillus* spp., *Bacillus* spp., and *Pseudomonas* spp. [7]. However, lipase production by lactic acid bacteria (LAB) has not been extensively studied. LAB are widely regarded as Generally Recognized as Safe (GRAS) [8] and are known for their various probiotic properties [9]. This has led to growing interest in LAB as potential lipase producers. LAB have a long history of use in fermented foods, including dairy products [10], meat [11], and sausages [12]. The ability of LAB to hydrolyze triglycerides through lipase production results in the release of free fatty acids, which significantly enhances the nutritional and flavor profiles of the products [13]. Therefore, screening lactic acid bacteria with high lipase activity is of considerable importance for the food industry.

*Lactiplantibacillus plantarum* (*L. plantarum*) is an important species of LAB due to its widespread distribution in various fermented foods [14]. Compared to other LABs, *L. plantarum* is considered a superior source of lipase and esterase enzymes [1]. Early studies have demonstrated that *L. plantarum* contains different types of lipases and esterases, which play a crucial role in lipid metabolism [3,15]. Historically, lipases were primarily analyzed through isolation and purification to elucidate their structures and properties [16]. In recent years, genomic analysis has proven to be an effective means for rapid identification of lipases/esterases [17]. Wang et al. [18] identified lipase-related genes through genome mining technology, revealing the key roles of these genes in lipid metabolism. Furthermore, Park et al. [19] employed whole-genome sequencing to elucidate the lipase gene of *Malassezia restricta* KCTC 27527. However, research on the safety and probiotic properties of high lipase-producing LAB strains remains limited.

The function of lipase varies significantly across different bacterial strains. For example, lipase acts as an important virulence factor in *Staphylococcus aureus* (*S. aureus*) and functions as a bioactive compound in LAB strains [20,21]. Limited research has been conducted on the safety and beneficial properties of high lipase-producing LAB strains. Not all LAB strains are reported to be safe. Therefore, the safety of lactic acid bacteria is getting increasing attention. Recent studies have demonstrated that evaluating the safety and probiotic properties of LAB is a prerequisite for their use as probiotics in food [22]. However, due to the diverse sources and growth environments of different LAB strains, further investigation is required to assess the safety and probiotic properties of the target strains [23]. With the rapid development of whole-genome sequencing technology, genomics has become an essential tool for uncovering the functional and safety characteristics [24], metabolic pathways [25], genetic backgrounds [25], and stress response mechanisms [26] of LAB. For instance, a study by Ahmed et al. [27] reveals safety-related genes in *Enterococcus lactis* strains through a comprehensive genome-wide analysis. In addition, Jiang et al. [28] assessed the safety and probiotic properties of *Lactobacillus salivarius* CGMCC20700 based on whole-genome sequencing. Therefore, it is important to investigate the genes associated with lipase activity, safety characteristics, and probiotic properties in LABs.

The LAB exhibiting lipolytic activity hold significant potential in food biotechnology. In this study, *L. plantarum* B22 was isolated from goat’s milk and demonstrated a remarkably high esterase/lipase activity. Crucially, the genetic determinants (esterase/lipase-encoding genes), enzyme structural features, enzyme substrate binding mechanisms, beneficial probiotic properties, and safety profile of this strain remain unexplored. To address this gap, we employed integrated whole-genome sequencing and phenotypic analyses to systematically investigate the relationship between key phenotypes of *L. plantarum* B22 and its underlying genomic features. The specific objectives of this study were as follows: (1) to identify and characterize the genetic determinants and structural features responsible for the high lipolytic activity; (2) to elucidate the substrate binding mechanisms of key esterases/lipases using molecular docking; and (3) to comprehensively assess the probiotic traits and safety profile. This study provides genomic insights for predicting the function of target strains.

## 2. Materials and Methods

### 2.1. Materials and Strains

The chemicals and reagents utilized in this research are as follows: de Man-Rogosa-Sharpe (MRS) agar and broth, olive oil, bile salts, pepsin, trypsin, triton X-100, and p-nitrophenyl acetate (p-NPA) were provided from Sigma–Aldrich (St. Louis, MO, USA). All the antibiotic susceptibility disks (diameter: 2 mm) were purchased from Changde Bickerman Biotechnology Co., Ltd. (Beijing, China). Gram staining was carried out using the Gram Stain Kit (Solarbio, Beijing, China).

### 2.2. Strains and Growth Conditions

These 156 LAB strains were isolated from Yunnan specialty resources, stored in Yunnan (Appendix A), and preserved in the Key Laboratory of Food Processing and Safety Control, Yunnan Agricultural University, Kunming, China. LAB strains were cultured in MRS broth (pH 6.5) at 37 °C for 24 h with an inoculation concentration of 1% (*v*/*v*). The strain was subcultured for three consecutive generations for preservation. *Staphylococcus aureus* ATCC 25923 was purchased from the American Type Culture Collection (ATCC, Manassas, VA, USA). *Escherichia coli* CICC 10389 was purchased from the China Center of Industrial Culture Collection (CICC, Beijing, China). *Salmonella* WX29 and *Pseudomonas aeruginosa* ST were independently isolated by the laboratory and preserved in the Key Laboratory of Food Processing and Safety Control, Yunnan Agricultural University, Kunming, China.

### 2.3. Screening Esterases/Lipases-Producing Strains

A total of 156 LAB strains stored in the laboratory at −80 °C were tested using a plate-based screening method. For preliminary screening, 20 μL of the fermentation broth of LAB cultured to the logarithmic phase was spread onto neutral red oil decomposition agar plates (composed of 10 g/L peptone, 5 g/L beef extract, 5 g/L NaCl, 10 g/L olive oil, 17 g/L agar, and 1 mL/L of 1.6% (*w*/*v*) neutral red aqueous solution, final pH 7.2), which were then incubated at 37 °C for 24 h. The appearance of red-colored colonies was observed to identify potential esterase/lipase producers. A 0.8 cm hole was punched in a glycerol tributyrate agar (pH 7.0) plate containing tryptone (2.5 g/L), casein peptone (2.5 g/L), yeast extract (3 g/L), NaCl (5 g/L), glycerol tributyrate (1%, *v*/*v*), and agar (17 g/L). Then, 100 μL of the homogeneous LAB fermentation broth was aspirated into the hole, and the plate was incubated until the broth reached the logarithmic phase. The plates were incubated at 37 °C for 24 h using a caliper (Deli Co., Ningbo, Zhejiang, China) to measure the size of the clear zone (mm) [29].

Extracellular esterase/lipase activity was measured using p-nitrophenyl palmitate (p-NPP) as a substrate. A total of 200 μL of p-NPP substrate solution was mixed with 2.1 mL of Tris-HCl buffer and preheated at 40 °C for 5 min. Subsequently, 200 μL of the enzyme solution was added, and the reaction was carried out for 10 min. Then, 1 mL of 0.5 mol/L trichloroacetic acid was added to stop the reaction, and the mixture was left for 5 min. Afterward, 3 mL of 0.5 mol/L NaOH was added to adjust the pH. A blank was prepared by replacing 200 μL of enzyme solution with 200 μL of distilled water under identical conditions. The absorbance at 410 nm was measured, and the concentration of p-nitrophenol (p-NP) generated was calculated using the standard curve. The enzyme activity was then determined. One unit of enzyme activity (U) was defined as the amount of enzyme that releases 1 μmol of p-nitrophenol per minute under specified conditions [1].Esterases/Lipases activity (U/mL) = cV/tV′
where c is the concentration of p-nitrophenol (μmol/L); V is the final volume of the reaction solution after acid-base adjustment (mL); V′ is the volume of the enzyme solution used (mL); t is the incubation time (min).

### 2.4. Genome Sequencing and Analysis

#### 2.4.1. DNA Extraction of *L. plantarum* B22

*Lactiplantibacillus plantarum* B22 was cultured in 150 mL MRS medium for 14 h, followed by centrifugation at 8000 g and 4 °C for 5 min. After centrifugation, the pellet was ground in liquid nitrogen and lysed with 0.5% SDS lysis buffer containing proteinase K (0.4 mg/mL) and 0.5% mercaptoethanol. After lysis was completed and the mixture cooled to room temperature, the supernatant was centrifuged and extracted twice with chloroform/isoamyl alcohol (24:1). DNA was then precipitated by the addition of isopropanol, followed by gentle mixing and centrifugation at 8000× *g* for 10 min at 4 °C. The DNA was initially purified using an OMEGA purification column (Omega Bio-Tek, Norcross, GA, USA), followed by further purification with Ampure XP beads. Finally, the DNA was assessed for quality using a NanoDrop spectrophotometer (NanoDrop Technologies, Inc., Wilmington, DE, USA) at a wavelength of 260 nm and 280 nm to measure the absorbance and calculate the A260/A280 ratio, and using a Qubit fluorometer (Life Technologies, Inc., Carlsbad, CA, USA) with the Qubit dsDNA BR Assay Kit (Fisher Scientific, Waltham, MA, USA) to determine the concentration [30].

#### 2.4.2. Library Construction

The amount of 2.5 μg of quality-controlled DNA was extracted, magnetic beads were added for purification, and 1 μL of the sample was quantified using a Qubit fluorometer. DNA fragments were mechanically fragmented by sonication, followed by purification, end-repair, 3′ end addition, and ligation for sequencing. Fragment size selection was performed using agarose gel electrophoresis, followed by PCR amplification to generate the sequencing library using the NEBNext^®^ Ultra™ DNA Library Preparation Kit (New England, Ipswich, MA, USA). The constructed libraries were then subjected to quality control. The libraries were then loaded onto R9.4 sequencing chips and sequenced using a PromethION sequencer (Oxford Nanopore Technologies, Oxford, UK) for 48–72 h [30].

#### 2.4.3. Gene Annotation and Analysis

Sequence similarity analysis was performed using NCBI BLAST (version 1.4.0) (http://blast.ncbi.nlm.nih.gov/, accessed on 1 September 2024). The Cluster of Orthologous Groups of proteins (COG) database was used for functional annotation of genes involved in carbohydrate transport and metabolism in *L. plantarum* B22. Metabolic pathways in the genome of *L. plantarum* B22 were annotated using the KEGG database. Putative genes involved in antimicrobial compound production in the genome of *L. plantarum* B22 were identified using antiSMASH 5.0. Virulence factor genes in the *L. plantarum* B22 genome were annotated using the Virulence Factor Database (VFDB). The screening criteria were similarity > 50% and E-value < 1 × 10^−10^. Genes associated with antibiotic resistance in the *L. plantarum* B22 genome were predicted using the CARD (Comprehensive Antibiotic Resistance Database). The criteria used were E-value < 1 × 10^−2^, coverage > 70%, and similarity > 30% [31].

#### 2.4.4. Prediction and Analysis of Esterase/Lipase Genes in *L. plantarum* B22 Genome

The gene sets obtained from the sequencing of *L. plantarum* B22 were annotated in 13 databases, including NR, Swissprot, Pfam, NOG, GO, KEGG, CAZy, VFDB, CARD, PHI, ResFinde, TCDB, and QSP. The genes with the function of esterases/lipases were screened through comparing parameters with functional annotations, such as Identity, E_value, and Score [32].

#### 2.4.5. Unraveling the Structural Features and Substrate Binding Mechanisms of Esterases/Lipases

The experiment was performed in accordance with previously reported methods with slight modifications [33,34]. Primary structure analysis (e.g., molecular weight, isoelectric point) and physicochemical properties of esterases/lipases were predicted using the ExPASy ProtParam Server (http://web.expasy.org/protparam/, accessed on 28 April 2025). Secondary structure prediction was conducted via the GOR IV method on the NPS@ server (http://npsa-pbil.ibcp.fr/cgi-bin/npsa_automat.pl?page=/NPSA/npsa_gor4.html, accessed on 28 April 2025). Transmembrane helices were identified using TMHMM 2.0 (http://www.cbs.dtu.dk/services/TMHMM/, accessed on 28 April 2025). Phylogenetic analysis was performed in MEGA 4.1 using the neighbor-joining method (1000 bootstrap replicates). For molecular docking, ligand structures (e.g., triglycerides) were retrieved from PubChem (https://pubchem.ncbi.nlm.nih.gov/, accessed on 30 April 2025), converted to 3D conformations in ChemOffice 20.0, and saved as .mol2 files. Protein structures were obtained from the RCSB PDB database (https://www.rcsb.org/, accessed on 30 April 2025), preprocessed in PyMOL 2.6.0 (removing water molecules and cofactors), and saved as .pdb files. Energy minimization and active pocket identification were conducted in MOE 2019 (Chemical Computing Group). Docking simulations were performed with 50 independent runs, and binding affinities were ranked by calculated binding energy (ΔG). Visualization of ligand–receptor interactions was achieved using PyMOL 2.6.0 and Discovery Studio 2019.

### 2.5. Methodology for Probiotic Properties Evaluation of L. plantarum B22

#### 2.5.1. Acid and Bile Resistance of *L. plantarum* B22

The acid resistance of the strain was determined using the viable plate counting method. Logarithmic-phase *L. plantarum* B22 was harvested by centrifugation at 8000× *g* for 5 min at 4 °C. The optical density at 600 nm was adjusted to 0.5 ± 0.02 using 10 mM sterile phosphate buffer (pH 7.0). Subsequently, 1.0 mL of the OD-adjusted bacterial suspension was added to MRS medium (9 mL, pH 2.0 and 3.0) and incubated at 37 °C for 3 h. The viable bacteria were quantified by the plate counting method at 0 h and 3 h, and the survival rate was calculated [35].

The bile salt tolerance of *L. plantarum* B22 was determined according to the method reported by Wei et al. [36]. The bile salt tolerance of the strains was evaluated by adding 1.0 mL of the *L. plantarum* B22 bacterial suspension to MRS medium (9 mL, pH 6.5) containing 0.3% (*w*/*v*) bile. The cultures were incubated at 37 °C for 3 h, and the viable bacteria were quantified at 0 h and 3 h using the plate counting method to calculate the survival rate.

#### 2.5.2. Gastrointestinal Tract Tolerance

The tolerance of *L. plantarum* B22 to simulated gastrointestinal conditions was assessed according to the method reported by Fei et al. [37] with minor modifications. Gastric juice was prepared with phosphate-buffered saline (PBS, pH 3.0) and supplemented with pepsin (5 mg/mL; Sigma, USA). Intestinal fluid was prepared using PBS (pH 7.4) containing 0.1% (*w*/*v*) trypsin(0.1 mg/mL; Sigma, St. Louis, MO, USA) and 0.3% (*w*/*v*) bile salts (Sigma, St. Louis, MO, USA). Logarithmic-phase bacterial cells were collected by centrifugation at 8000× *g* at 4 °C for 5 min and resuspended in simulated gastric fluid to a final concentration of approximately 10^8^ CFU/mL. The suspension was incubated at 37 °C for 2 h, and survival was determined by plate counting. The strains treated with simulated gastric fluid were then transferred to simulated intestinal fluid and incubated for an additional 2 h. The survival rate was determined using the plate counting method.

#### 2.5.3. Assays for Antioxidant Activities

The *L. plantarum* B22 cell-free supernatant (CFS) was prepared according to the method described by Wang et al. [38] with some modifications. The scavenging ability of 2,2-diphenyl-1-picrylhydrazyl (DPPH) radicals was evaluated following the modified procedure of Yang et al. [39]. In this assay, 2 mL of a 0.4 mM DPPH solution in methanol was mixed with either 2 mL of distilled water (as the control) or the *L. plantarum* B22 CFS. The mixtures were incubated at 25 °C for 30 min in the absence of light. After incubation, the absorbance at 517 nm was measured, and the radical scavenging activity was calculated using the following formula:DPPH radical scavenging rate (%) = [(A_0_ − As)/A_0_] × 100
where A_0_ represents the absorbance of distilled water plus the DPPH ethanol solution, and As represents the absorbance of the sample solution plus the DPPH ethanol solution.

For the ABTS radical scavenging test, the method described by Yang et al. [39] was adapted with minor modifications. A working solution of ABTS was prepared by combining 5 mM potassium persulfate with 7 mM ABTS, which was then diluted with a 20 mM sodium phosphate buffer (pH 7.4) to achieve an absorbance of 0.7 ± 0.02 at 734 nm. In the assay, 150 μL of either the *L. plantarum* B22 CFS or distilled water (control) was mixed with 150 μL of the ABTS solution and incubated at 37 °C for 10 min. The absorbance was then measured at 734 nm, and the scavenging activity was computed using the following formula:ABTS + radical scavenging rate (%) = [(A_0_ − A_1_)/A_0_] × 100
where A_0_ is the absorbance of the blank control, and A_1_ is the absorbance of the sample with the ABTS+ working solution.

#### 2.5.4. Evaluation of Antibacterial Activity

The *L. plantarum* B22 CFS was prepared according to the method described by Wang et al. [38] with some modifications. The Oxford cup plate method was employed to determine the bacteriostatic activity of *L. plantarum* B22. *Staphylococcus aureus* ATCC 25923, *Escherichia coli* CICC 10389, *Salmonella* WX29, and *Pseudomonas aeruginosa* ST were selected as indicator bacteria. The indicator bacteria were inoculated into Luria Bertani (LB) medium (pH 7.2) with an inoculum volume fraction of 1%. Using the Oxford cup double-layer plate method, 200 µL of the *L. plantarum* B22 CFS was added to each well. The plates were allowed to diffuse at room temperature for 4 h before being incubated at 37 °C for 12 h. The diameter of the bacterial inhibition zone was then measured to assess the bacteriostatic activity of the *L. plantarum* B22 CFS.

### 2.6. Safety Analysis of L. plantarum B22

#### 2.6.1. Detection of Harmful Metabolites of *L. plantarum* B22

(1)Hemolysis experiment

Following a modified version of Lu et al. [40]. protocol, activated *L. plantarum* B22 was inoculated onto Columbia blood agar plates (Thermo Fisher Scientific, Oxoid, UK, pH 7.3) and incubated at 37 °C for 48 h. The presence of hemolytic activity was assessed by observing hemolysis around *L. plantarum* B22 colonies. β-Hemolytic *S. aureus* ATCC 25923 was used as a control.

(2)Amino Acid Decarboxylase Test

The amino acid decarboxylase test was performed according to the method described by Li et al. [41] with some modifications. The activated *L. plantarum* B22 was inoculated in the amino acid decarboxylase induction medium containing arginine, lysine, and ornithine decarboxylase biochemical tubes (pH 6.0), respectively, and cultured at 37 °C for 24 h. Biochemical tubes without *L. plantarum* B22 inoculation served as a control. The tubes were sealed with liquid paraffin and incubated at 37 °C for 24 h. A color change in the test tube to purple indicated a positive result, while yellow indicated a negative result.

#### 2.6.2. Antibiotic Sensitivity Analysis of *L. plantarum* B22

The antibiotic drug sensitivity test was performed according to the disk diffusion method, where a 10^8^ CFU/mL culture of *L. plantarum* B22 (200 µL) was spread evenly onto MRS agar (pH 6.5). The antibiotic drug sensitivity test piece was then placed in the plate using flame-sterilized forceps. The plates were incubated overnight at 37 °C, and the diameter of the circle of inhibition (mm) was determined. The diameter of the inhibition zone surrounding the disk was measured and categorized as sensitive (S; >20.5 mm), intermediate (I; 10.5–20.5 mm), and resistant (R; <10.5 mm), following the method described by Zareie et al. [42].

### 2.7. Statistical Analysis

All statistical analyses were performed using SPSS 23.0 software (IBM SPSS Statistics). Data are expressed as mean ± SD standard deviation (n = 3). All data were subjected to one-way analysis of variance (ANOVA) and Duncan’s new multiple range test. Normality of data distribution was assessed using the Shapiro–Wilk test, and the homogeneity of variance was assessed with Levene’s test. Results were considered statistically significant at *p*-value < 0.05.

## 3. Results and Discussion

### 3.1. Screening and Identification of High-Esterase/Lipase-Producing LAB Strains

#### 3.1.1. Screening of High-Esterase/Lipase-Producing LAB Strains

To identify LAB strains with a high esterase/lipase activity, a total of 156 strains preserved in the laboratory were screened using an agar plate assay. As shown in Figure 1A,B, strains B22 and B61 exhibited significant color changes on neutral red plates and formed clear hydrolytic halos on tributyrin agar plates, indicating their ability to hydrolyze tributyrin, thus demonstrating a strong esterase/lipase activity. Tributyrin, a widely used substrate for screening lipase- and esterase-producing microorganisms, enables effective evaluation of lipase activity by observing hydrolysis zones around colonies grown on nutrient agar containing tributyrin [43,44]. Similar methodologies have been applied in previous studies to screen for lipase-producing microorganisms. For example, Salwoom et al. [29] successfully identified, through tributyrin agar plate screening, a lipase-producing strain of *Pseudomonas* sp. LSK25, which exhibited a remarkable enzymatic activity. These findings highlight the utility of tributyrin agar as a robust tool not only for esterase/lipase screening but also for identifying microorganisms with high enzymatic activities. Subsequent esterase/lipase activity assays revealed that strain B22 exhibited the highest enzymatic activity (3.45 ± 0.13 U/mL) and was selected as the target strain for further experiments (Figure 1C). These results suggest that strain B22 is a promising candidate for esterase/lipase production, with significant potential for future applications in biotechnological and industrial fields.

#### 3.1.2. Morphological and Molecular Identification of High Esterase/Lipase-Producing LAB Strain

Morphological characterization revealed that strain B22 formed round, medium-sized, convex, whitish, and moist colonies on MRS agar plates (Figure 1D). Optical microscopy analysis showed that strain B22 exhibited typical Gram-positive characteristics (Figure 1E), supporting its classification as a Gram-positive bacterium. To confirm the species identity of strain B22, phylogenetic analysis was performed based on the 16S rRNA gene sequence. The results indicated a high degree of similarity between strain B22 and the *Lactiplantibacillus plantarum* species, leading to its designation as *Lactiplantibacillus plantarum* B22 (Figure 1F). This identification aligns with the findings of Kumar et al. [45], who effectively identified unknown strains through a combination of morphological features and molecular methods, demonstrating the reliability of this approach for LAB species identification. Similarly, Jiang et al. [28] utilized morphological observations and phylogenetic tree analysis to successfully identify a strain as *Lactobacillus salivarius*. These studies underscore the effectiveness of combining morphological characteristics with molecular biology techniques, particularly 16S rRNA gene analysis, as a robust strategy for accurate species identification. Accurate species identification not only establishes a foundation for microbial taxonomy but also provides a theoretical basis for exploring genetic traits and probiotic functional factors [46].

### 3.2. Elucidation of the Structure and Molecular Mechanism of Substrate Interactions of Esterase/Lipase by Genome-Wide Annotation and Molecular Docking

#### 3.2.1. Genomic Characteristics and Functional Annotation of *L. plantarum* B22

The whole genome of *L. plantarum* B22 was sequenced using Illumina HiSeq and PacBio platforms, providing high-quality data for detailed analysis. The genome of *L. plantarum* B22 was assembled into a complete circular structure with a total length of 3,290,520 base pairs (bp) and an average GC content of 47.53% (Figure 2A). GC content is a key genomic feature that reflects nucleotide composition and serves as an indicator of bacterial evolution [47]. It also provides insights into environmental adaptability and metabolic diversity [48]. The relatively high GC content of *L. plantarum* B22 suggests its stability. This genomic stability makes *L. plantarum* B22 a promising candidate for long-term use in fermented foods. Genome annotations are shown in Table 1, identifying 2830 coding genes with an average length of 884 bp, covering 84.49% of the genome. Additionally, 16 rRNA genes, 69 tRNA genes, and 42 sRNA genes were detected. These non-coding RNAs play crucial roles in protein synthesis, gene regulation, and environmental adaptation [49]. Among the tRNA genes, 21 amino acids were annotated, with glycine tRNA genes being the most abundant (6 copies) and cysteine tRNA genes the least abundant (1 copy). Notably, glycine contributes a sweet flavor to fermented products, enhancing consumer acceptance and preference [50]. Whole-genome sequencing provides crucial support for the in-depth exploration of the genetic foundation of *L. plantarum* B22. It also lays a solid foundation for uncovering the potential of LAB in food fermentation, probiotic functionalities, and biotechnological applications [51].

A total of 2774 genes in *L. plantarum* B22 were annotated in the COG database, with a sum of 23 subcategories, as shown in Figure 2B. It was divided into four major categories related to cell processes and signaling (653), information storage and processing (619), metabolism (1160), and poorly characterized (342). The majority of genes (284) were devoted to carbohydrate transport and metabolism, while those related to translation ranked second (281). In addition, lipid transport and metabolism (108) were also noted. The number of genes is higher than that in ref. [41], indicating that *L. plantarum* B22 has the potential for lipid transport and metabolism. KEGG database annotation showed that 2444 genes of *L. plantarum* B22 were annotated, with 109, 273, 184, 121, 1711, and 46 genes involved in cellular process, environmental information processing, genetic information processing, human diseases, metabolism, and organism systems, respectively (Figure 2C). Furthermore, the largest proportion of genes belonged to metabolism, including 288 genes participating in lipid metabolism. Studies have shown that esterase/lipase is a key enzyme in lipid metabolism [52].

#### 3.2.2. Identification of Esterase/Lipase Genes and Bioinformatics Analysis

Genomic analysis of *L. plantarum* B22 identified 13 genes encoding the esterase/lipase activity. Subsequently, phylogenetic analysis was performed comparing these esterases/lipases with known lipases (TAT54020.1, AZU39759.1, AQY70037.1) and esterases (VDH12635.1, VDH11235.1, BBAB1118.1) to elucidate their evolutionary relationships. As illustrated in Figure 3A, the esterase/lipase genes of *L. plantarum* B22 clustered primarily into four distinct phylogenetic groups. Notably, genes 3060, 2553, and 0817 exhibited high sequence similarity to the characterized lipase TAT54020.1. Significantly, gene3060 was found to be identical to the known lipase TAT54020.1, gene2576 was found to be identical to the known esterase VDH12635.1, and gene0798 was identical to the known esterase BBAB1118.1. This demonstrates that our method is appropriate for the identification of esterases/lipases. Further, the ProtParam tool (http://web.expasy.org/protparam/, accessed on 28 April 2025) from the ExPASy database was used to predict and analyze the basic physicochemical properties of the esterases/lipases. The information on these esterase/lipase genes is listed in Table 2. The primary structure of proteins is fundamental for studying their structure, physiological functions, and mechanisms of action [53]. As shown in Table 2, except for gene1511, gene1598, and gene2581, all other esterases/lipases are acidic. This finding is consistent with that of Califano et al. [54], who reported that microbial lipases typically have acidic isoelectric points, with the lipase from *Candida rugosa* having an isoelectric point of 4.5. Additionally, most instability indices were less than 40, indicating that these proteins are stable. The aliphatic indices and grand average of hydropathy (GRAVY) values were similar to those reported by Liao et al. [55], suggesting that these lipases are lipophilic proteins.

The secondary structure of the lipases was predicted using the GOR IV method on the NPS@ online server. The results are shown in Table 2 and Appendix A. Specifically, h (blue) represents α-helices, e (red) represents β-extended strands, and c (pink) represents random coils. Different mRNAs encode different secondary structures. Slow translation tends to encode β-sheets and random coils, while fast translation favors α-helices. As shown in the table, most of the annotated lipase mRNA was translated at a slower rate. The presence of transmembrane regions was predicted using the TMHMM online tool. As shown in Appendix A, no transmembrane structure exists except for gene1598, and gene2581. The lack of intersection between the two lines indicates that the annotated lipases/esterases are not located in transmembrane regions, which is consistent with the findings of Liu et al. [56]. The results of multiple sequence alignment are shown in Appendix A. The genes 3060, 3059, 2553, and 0798 possess the typical catalytic triad (Ser-Ala-Gly/His) of lipases/esterases. Future studies will functionally validate *L. plantarum* B22 lipase genes through heterologous expression.

#### 3.2.3. Molecular Mechanism of Substrate Interactions of Esterase/Lipase

Further molecular docking analysis was performed to investigate the interactions between gene3060, gene3059, gene2553, and gene0798 with tributyrin using MOE 2019 software (Figure 3B–E). Generally, a docking energy value of less than −4.25 kcal/mol indicates some binding activity, less than −5.0 kcal/mol indicates a good binding activity, and less than −7.0 kcal/mol indicates a strong binding activity. The docking results showed that all seven lipases exhibited a good binding activity with tributyrin, with docking energies ranging from −5.528 to −6.7784 kcal/mol. This indicates that the annotated lipases can bind to tributyrin, consistent with the findings presented in Figure 1B. The low docking energy of tributyrin–gene0798 (−6.7784 kcal/mol) suggests that the enzyme encoded by gene0798 may have a stronger binding activity with tributyrin. The enzyme encoded by gene0798 forms hydrogen bonds with tributyrin through Thr60, Ser58, and Gly107, while residues Ala314, Tyr202, Pro317, Phe285, His313, Met245, and Leu241 interact hydrophobically with the compound. These genes highlight the potential of *L. plantarum* B22 for esterase/lipase synthesis and binding to substrates, demonstrating a high esterase/lipase activity of *L. plantarum* B22. Similarly, previous genomic studies on spoilage-associated *Shewanella putrefaciens* have identified lipase-encoding genes, providing foundational insights into lipid biosynthesis [57].

### 3.3. Evaluation of Probiotic Properties of L. plantarum B22

#### 3.3.1. Acid and Bile Salt Tolerance

Acid and bile salt tolerance are considered key biomarkers for LAB [58]. In the human digestive system, gastric pH fluctuates, typically around 3.0, but can drop as low as 1.8 under fasting conditions. For probiotics to exert beneficial effects, they must withstand such extreme acidic environments and successfully transit through the stomach. As shown in Figure 4A,B, *L. plantarum* B22 demonstrated notable acid and bile salt tolerance. Under acidic conditions (pH 3.0), the viable cell count and survival rate of the strain were 8.16 ± 0.04 log CFU/mL and 52.24 ± 4.83%, respectively. At pH 2.0, the viable cell count and survival rate were 7.97 ± 0.02 log CFU/mL and 33.94 ± 1.98%, respectively. These findings suggest that *L. plantarum* B22 possesses robust acid resistance. Genome analysis identified 2 genes potentially encoding Na+/H+ antiporters (Table 2). The Na+ /H+ antiporters play a predominant role in maintaining intracellular pH [59]. Overly acidic environments can inhibit the growth of LAB [60]. In the presence of 0.1% bile salts, the strain exhibited a viable cell count of 8.34 ± 0.06 log CFU/mL and a survival rate of 76.19 ± 7.56%. Current research suggests that bile salt tolerance in LAB is closely related to their cyclopropane-fatty-acyl-phospholipid synthase [61]. In *L. plantarum* B22, two genes encoding cyclopropane-fatty-acyl-phospholipid synthase family protein were found. In addition, an annotation to 2 genes encoding AI-2E proteins in *L. plantarum* B22 (Table 3) was made. It was shown that the AI-2E protein was associated with the resistance of the strain to high salt stress [62]. Notably, the survival rates observed in this strain are comparable to those of *Lactiplantibacillus rhamnosus* B31-2 under similar conditions [63]. The ability to tolerate low pH and bile salts is a key factor for the survival of LAB during passage through the gastrointestinal tract [64]. This indicates that *L. plantarum* B22 could be an effective candidate for gastrointestinal tract colonization.

#### 3.3.2. Gastrointestinal Survival Rate

The survival of probiotics under simulated gastrointestinal (GI) conditions is a key indicator of their potential efficacy in the gut environment [65]. As shown in Figure 4C,D, the viable cell count of *L. plantarum* B22 decreased from 8.48 ± 0.01 log CFU/mL to 8.30 ± 0.03 log CFU/mL after 2 h of gastric juice treatment. Despite this decline, the cell count remained above 7 log CFU/mL, indicating that the strain retained substantial viability. However, the survival rate dropped significantly to 66.23 ± 4.03%, suggesting a suppressive effect of gastric juice on the strain. Similarly, the viable count decreased further after 2 h of intestinal juice treatment, from 8.30 ± 0.03 log CFU/mL to 8.10 ± 0.08 log CFU/mL. The survival rate significantly declined to 41.42 ± 7.22%. These findings are consistent with previous studies showing that *L. plantarum* DMDL 9010 maintained viability after simulated gastrointestinal digestion [66]. The observed GI survival ability of *L. plantarum* B22 may be attributed to stress-related genes identified in its genome (Table 3). These genes likely play a critical role in enabling the strain to withstand harsh conditions encountered in the gastrointestinal tract or during fermentation. For instance, genes involved in acid and bile tolerance, oxidative stress responses, and membrane integrity could enhance the strain’s resilience. Maintaining a viable cell count above 7 log CFU/mL under simulated GI conditions is crucial, as this threshold is considered essential for probiotics to confer health benefits. The results suggest that *L. plantarum* B22 could potentially survive gastrointestinal transit and exert beneficial effects in the host gut environment. Further studies exploring the mechanisms behind its stress tolerance and functional contributions in vivo are warranted.

#### 3.3.3. Antioxidant Activity Analysis

Probiotics can function as antioxidants, contributing to the maintenance of redox balance in the gut. The scavenging activities of hydroxyl radicals, superoxide anions, and DPPH free radicals by the fermentation supernatant and bacterial precipitate of *L. plantarum* B22 are shown in Figure 4E. The results demonstrate that *L. plantarum* B22 exhibits notable antioxidant activity, with the CFS showing higher radical-scavenging capacity than the bacterial precipitate. This difference may be attributed to the structure and composition of metabolites secreted by the strain, as well as the presence of oxidative stress-related genes. The genome of *L. plantarum* B22 harbors genes encoding key antioxidant enzymes, including catalase, glutathione peroxidase, thioredoxin, thioredoxin reductase, and thiol peroxidase (Table 3). Catalase plays a direct role in detoxifying hydrogen peroxide and reactive oxygen species (ROS) [67]. Glutathione peroxidase, a critical enzyme in maintaining glutathione homeostasis, reduces disulfide bonds in glutathione and eliminates free radicals [68]. In addition, the genome contains multiple genes associated with manganese (Mn) transport, including manganese transport proteins, Mn ABC transporters, and Mn-dependent inorganic pyrophosphatase (Table 3). The Mn accumulation system regulates Mn transport and accumulation, enhancing the strain’s tolerance to oxidative stress [69]. Manganese ions (Mn^2+^) are known to protect against ROS by functioning as cofactors for superoxide dismutase and other enzymes involved in redox regulation. These findings suggest that *L. plantarum* B22 is a probiotic with strong antioxidant potential.

#### 3.3.4. Antimicrobial Properties

The antimicrobial activity of probiotics is of great importance in food production and preservation, as LAB can secrete a variety of metabolites with bacteriostatic effects, making them promising natural preservatives. The antimicrobial activity of *L. plantarum* B22 against three indicator strains was evaluated using the Oxford cup assay. As shown in Figure 4F, *L. plantarum* B22 demonstrated significant inhibitory effects against *Pseudomonas aeruginosa*, *Escherichia coli*, *Salmonella enterica*, and *Staphylococcus aureus*, with varying degrees of effectiveness. Among these, the highest inhibition was observed against *S. aureus*, with an inhibition zone of 17.70 ± 0.42 mm. This was followed by *Salmonella enterica* with an inhibition zone of 16.04 ± 1.08 mm. Previous studies have also reported similar findings. For instance, *L. plantarum* LR-14, isolated from Sichuan pickles, exhibited antimicrobial activity against *S. aureus* [70]. LAB are known to inhibit harmful microorganisms by producing bacteriocins, a class of ribosomally synthesized antimicrobial peptides [71]. Genomic analysis of *L. plantarum* B22 revealed the presence of four genes encoding two-peptide bacteriocins (Table 3), which are likely contributors to its antimicrobial activity. In addition to bacteriocins, organic acids such as lactic acid, citric acid, isobutyric acid, and acetic acid are key antimicrobial metabolites produced by LAB. These acids lower the surrounding pH, creating an unfavorable environment for pathogenic bacteria and leading to their death [72]. The production of such acids further underscores the potential of *L. plantarum* B22 as a natural antimicrobial agent. The identification of bacteriocin biosynthesis-related genes provides valuable insights into the genetic basis of antimicrobial activity in *L. plantarum* B22.

#### 3.3.5. Reveals Key Metabolism Pathways in *L. plantarum* B22

The KEGG pathway annotation identified 76 genes mapped to 7 lipid metabolism pathways (Appendix A). Among these, 22 genes were linked to fatty acid biosynthesis, 20 to glycerolipid metabolism, 19 to glycerophospholipid metabolism, 8 to fatty acid degradation, 5 to sphingolipid metabolism, and 1 each to primary and secondary bile acid biosynthesis.

A notable feature of *L. plantarum* B22 is its complete fatty acid biosynthesis pathway (Figure 5). Fatty acid synthesis is critical for producing essential cellular components and serves as a key energy reservoir for the cell [73]. The process begins with acetyl-CoA, which is converted into malonyl-CoA by acetyl-CoA carboxylase (EC:6.4.1.2). This reaction is a pivotal, rate-limiting step in fatty acid biosynthesis. Malonyl-CoA is subsequently converted to malonyl-ACP by malonyl-CoA-ACP transacylase (FabD). The pathway proceeds with fabH-encoded β-ketoacyl-ACP synthase catalyzing the formation of acetoacetyl-ACP, which undergoes reduction, dehydration, and further reduction through FabG, FabZ, and FabI (or FabK), respectively. These steps progressively elongate the carbon chain, cycling repeatedly to produce fatty acids such as caprylic acid, capric acid, lauric acid, myristic acid, palmitic acid, and stearic acid. The importance of EC:6.4.1.2 as a rate-limiting enzyme in the initial step of fatty acid synthesis is well documented [74]. Additionally, FabI and FabG are indispensable enzymes in type II fatty acid biosynthesis, playing a central role as catalytic controllers [75]. The robust expression of these genes in *L. plantarum* B22 suggests a strong metabolic capacity for synthesizing diverse fatty acids. In glycerolipid metabolism, glycerol is converted into phosphatidate through the action of pathway-specific enzymes encoded in the genome (Figure 5). Similarly, in glycerophospholipid metabolism, glycerone phosphate and CDP-glycerol are transformed into cardiolipin, a key phospholipid in bacterial membranes. Interestingly, the cls gene encoding cardiolipin synthase was annotated, consistent with findings in other bacterial species, such as *Devriesea agamarum* IMP2T [76]. These genomic annotations highlight the comprehensive lipid metabolism potential of *L. plantarum* B22. The strain’s capacity to synthesize fatty acids, glycerolipids, and phospholipids underscores its potential applications in functional food production, particularly in enhancing lipid profiles and contributing to flavor development. Future research should investigate the regulatory mechanisms governing these pathways and explore their industrial relevance. This genomic insight provides a foundation for developing *L. plantarum* B22 as a functional ingredient in lipid-enriched or fermented dairy products.

### 3.4. Safety Evaluation of L. plantarum B22

Lactic acid bacteria (LAB) are generally considered safe (GRAS) due to their long history of use in fermented foods. However, emerging research emphasizes the need for genomic and phenotypic assessments to confirm their safety [77]. The safety evaluation of LAB typically includes assessment of hemolytic activity, transferable antibiotic resistance genes, virulence factors, and undesirable metabolites [78].

#### 3.4.1. Hemolytic Activity Analysis

Hemolysis is a biological phenomenon caused by the production of hemolysins by bacteria during growth. It is commonly regarded as a key pathogenic factor for bacterial invasion of the host [79]. Hemolysins can disrupt red blood cells, leading to complete lysis and severe hemolytic reactions, which, in turn, can cause anemia and a range of other clinical symptoms [80]. Therefore, evaluating the hemolytic activity of LAB is an important aspect of assessing their safety as probiotics [81]. In this study, the hemolytic activity of *L. plantarum* B22 was tested on Columbia blood agar plates, with *S. aureus* ATCC 25923 (a known β-hemolytic strain) used as a positive control. As shown in Appendix A, no clear zone of lysis was observed around *L. plantarum* B22, unlike *S. aureus*, indicating that the strain does not produce β-hemolysin. This finding demonstrates that *L. plantarum* B22 lacks hemolytic pathogenic potential. Moreover, genomic analysis revealed no genes associated with hemolysis in the *L. plantarum* B22 genome, further confirming the absence of hemolytic activity at the molecular level. These results align with the study by Peres et al. [82], which also found that lactic acid bacteria generally do not exhibit hemolytic properties in safety evaluations. Overall, the hemolytic activity evaluation of *L. plantarum* B22 supports its safety for use in food and health applications.

#### 3.4.2. Analysis of Harmful Metabolites

Amino acid decarboxylases catalyze the decarboxylation of amino acids to produce biogenic amines, such as histamine, tyramine, putrescine, and cadaverine [83]. These biogenic amines are highly toxic and can pose health risks to humans. As a result, microorganisms capable of decarboxylating amino acids and producing biogenic amines are closely monitored and studied in food production [84]. As shown in Appendix A, *L. plantarum* B22 tested negative for amino acid decarboxylase activity. Crucially, whole-genome sequencing revealed no intact genes encoding tyrosine (*tdc*), arginine (*adc*), histidine (*hdc*), or lysine (*ldc*) decarboxylases, and no complete gene clusters for biosynthesizing tyramine, putrescine, histamine, or cadaverine. This genotype-phenotype concordance definitively establishes that *L. plantarum* B22 does not produce amino acid decarboxylases and, therefore, does not decompose amino acids to form biogenic amines. Consequently, the consumption of *L. plantarum* B22 does not pose a risk for biogenic amine-induced food poisoning.

#### 3.4.3. Antibiotic Resistance Testing

The widespread use of antibiotics across various fields has raised concerns regarding the antibiotic resistance of LAB [85]. Antibiotic resistance has become an important criterion for evaluating the suitability of bacterial strains as additives in human and animal feed. To assess the potential application of *L. plantarum* B22 in food products, this study evaluated its resistance to 20 common antibiotics (Table 4). The antibiotic sensitivity analysis followed the methodology outlined by Guan et al. [86]. As shown in Table 4, *L. plantarum* B22 exhibited sensitivity to a wide range of antibiotics, including penicillin, piperacillin, ampicillin, minocycline, cefuroxime, ceftazidime, ceftriaxone, cefoperazone, and erythromycin, with clear zone diameters of 27.60 ± 2.84 mm, 35.27 ± 0.33 mm, 31.89 ± 3.16 mm, 40.09 ± 3.46 mm, 41.55 ± 0.72 mm, 25.51 ± 2.80 mm, 39.43 ± 1.31 mm, 24.61 ± 0.99 mm, and 27.42 ± 1.46 mm, respectively. However, resistance was observed against several antibiotics, including streptomycin, amikacin, doxycycline, vancomycin, chloramphenicol, cefalexin, and cephazolin. Antibiotic resistance to one or more antibiotics is a common trait among many LAB strains [87]. The resistance to vancomycin, specifically, has been reported in various LAB strains, including LAB species used in fermented products [81]. In a study by Guo et al. [88], *Lactobacillus bulgaricus* strains isolated from traditional dairy products showed resistance to both streptomycin and vancomycin, consistent with our findings. Using the Antibiotic Resistance Database (CARD) for gene annotation, we identified resistance genes associated with 24 antibiotics, including tetracyclines, aminoglycosides, peptide antibiotics, and lincosamides (Appendix A). Furthermore, genomic analysis showed that these antibiotic resistance genes are located on the chromosome of *L. plantarum* B22, in line with findings by Ryser et al. [89]. This suggests that the resistance genes are relatively stable within the strain and are unlikely to spread through horizontal gene transfer, posing a minimal direct threat to human health.

#### 3.4.4. Virulence Factor Analysis

Virulence factors are molecules produced by pathogenic microorganisms that play key roles in the infection process of their host organisms [90]. Based on annotations from the Virulence Factor Database (VFDB) (Table 5), 50 potential virulence factors were identified in the genome of *L. plantarum* B22. In comparison, a total of 97 potential virulence factor genes were annotated in the genome of *Lacticaseibacillus rhamnosus* LR-ZB 1107-01 [91]. The majority of the virulence genes identified in both strains are related to stress survival, regulation, nutrition/metabolism, motility, immune modulation, biofilm formation, and adhesion. These virulence factors play crucial roles in the pathogenic processes of harmful microorganisms. However, it is important to note that the same genes found in LAB often serve different functions compared to pathogenic bacteria. In LAB, these genes are typically associated with beneficial effects, such as supporting probiotic functions. For example, certain adhesion-related genes, which are considered key virulence factors in pathogens [92], assist in the survival and colonization of beneficial bacteria in the gastrointestinal tract [93]. Similarly, biofilm and stress survival-related genes, which are associated with resistance and survival in pathogenic microbes [94], can have protective benefits for LAB. The biofilms produced by LAB can enhance survival under harsh conditions, such as during food processing, storage, and gastrointestinal transit [95]. *Staphylococcus aureus* and *Mycobacterium* species have high esterase and lipase activities and utilize these enzymes for invasive and virulence effects, such as disrupting host cell membranes and promoting tissue invasion [96]. In contrast, *L. plantarum*, while possessing esterase/lipase capabilities, primarily employs these enzymes to metabolize dietary lipids and generate flavor compounds [97]. Furthermore, *L. plantarum* has an extensive history of use, lacks genes for hemolytic toxins, and does not elicit adverse reactions in the host [98]. In summary, while LAB strains such as *L. plantarum* B22 may possess some of the same virulence factors as pathogenic microorganisms, the functional roles of these factors differ significantly.

## 4. Conclusions

In this study, *L. plantarum* B22 was identified as a high esterase/lipase-producing bacterium and comprehensively assessed using phenotypic and genomic approaches to evaluate its probiotic attributes and safety. Genomic analysis revealed the presence of 13 esterase/lipase genes, with 4 of them harboring conserved catalytic triads that are essential for enzymatic function. Molecular docking studies validated the interaction between esterase/lipase and tributyrin and are in accordance with the in vitro results. It is worth noting that gene0798 exhibited a high potential MolDock score of −6.7787 kcal/mol. Furthermore, phenotypic tests and genomic analysis proved that *L. plantarum* B22 possessed good antioxidant and antimicrobial abilities as well as good tolerance to the gastrointestinal environment. Importantly, the strain exhibited no antibiotic resistance, no hazardous metabolite production, and no virulence factors, thereby ensuring its safety for potential probiotic applications. Collectively, the probiotic functionality and safety of *L. plantarum* B22 were investigated through genomic and phenotypic analyses, providing a theoretical basis for the rapid genomic prediction of functional traits in unknown strains. Future studies will employ heterologous expression combined with X-ray crystallography and cryo-electron microscopy (cryo-EM) to elucidate the structure and catalytic mechanism of *L. plantarum* B22 esterase/lipase. Gene editing of key targets will further validate strain-specific probiotic functionality and safety.

## Figures and Tables

**Figure 1 foods-14-02354-f001:**
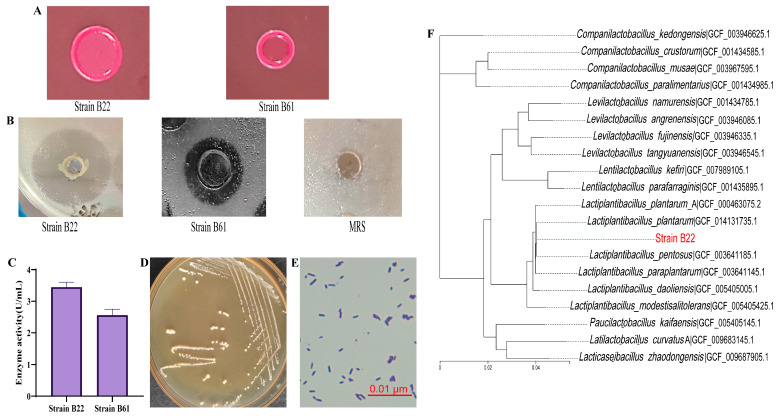
(**A**) A neutral red agar plate showing the growth of strain B22; (**B**) tributyrin agar plate showing the growth of strain B22; (**C**) esterase/lipase activity assay results for strain B22; (**D**) colony morphology of strain B22 on MRS agar; (**E**) the morphology of strain B22 observed under an optical microscopy; (**F**) a phylogenetic tree of strain B22.

**Figure 2 foods-14-02354-f002:**
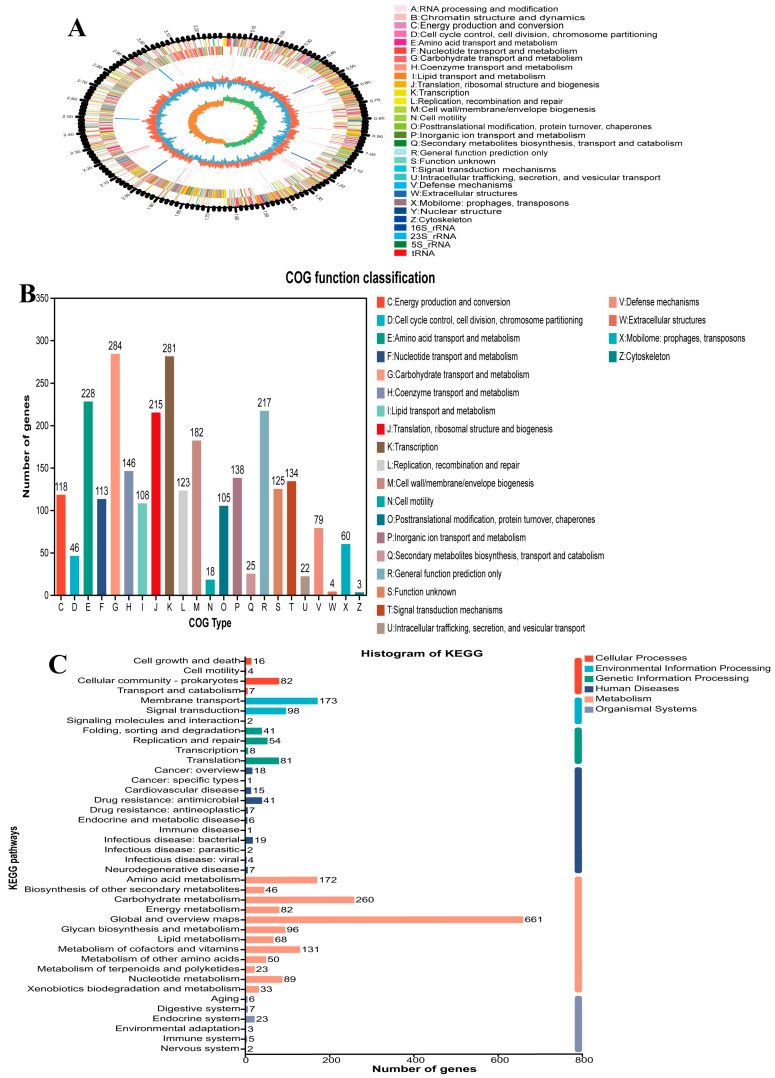
Genomic features and functional annotation of *L. plantarum* B22. (**A**) A genomic circle map of *L. plantarum* B22; (**B**) COG database annotation of *L. plantarum* B22; (**C**) KEGG database annotation of *L. plantarum* B22.

**Figure 3 foods-14-02354-f003:**
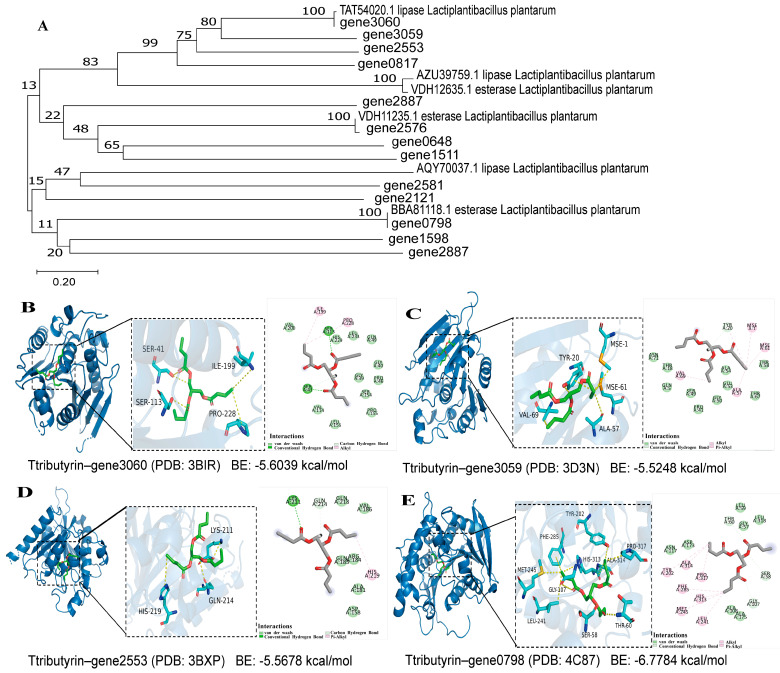
Evolutionary and structural analysis of esterase/lipase enzymes in *L. plantarum* B22. (**A**) Evolutionary tree of amino acid sequence of the esterase/lipase; (**B**–**E**) molecular docking of gene3060, gene3059, gene2553, and gene0798 with tributyrin.

**Figure 4 foods-14-02354-f004:**
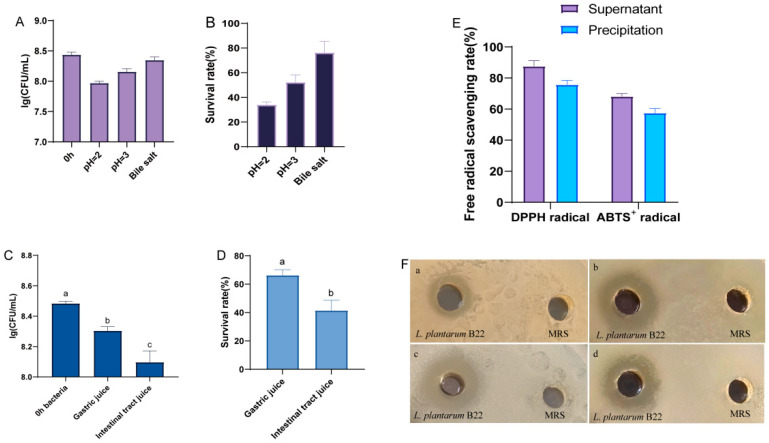
Stress tolerance and functional properties of *L. plantarum* B22. (**A**) Bacteria count of *L. plantarum* B22 at different pH and bile salt content; (**B**) a survival rate (%) of *L. plantarum* B22 at different pH and bile salt content; (**C**) bacteria count of *L. plantarum* B22 under simulated gastric and intestinal juice, different letters (a, b, c) represent significant differences ( *p* < 0.05); (**D**) a survival rate (%) of *L. plantarum* B22 under simulated gastric and intestinal juice, different letters (a, b) represent significant differences ( *p* < 0.05); (**E**) antioxidant capabilities of *L. plantarum* B22; (**F**) *L. plantarum* B22 inhibitory-circle. (**a**) *Pseudomonas aeruginosa*. (**b**) *Salmonella*. (**c**) *Staphylococcus aureus*. (**d**) *Escherichia coli*.

**Figure 5 foods-14-02354-f005:**
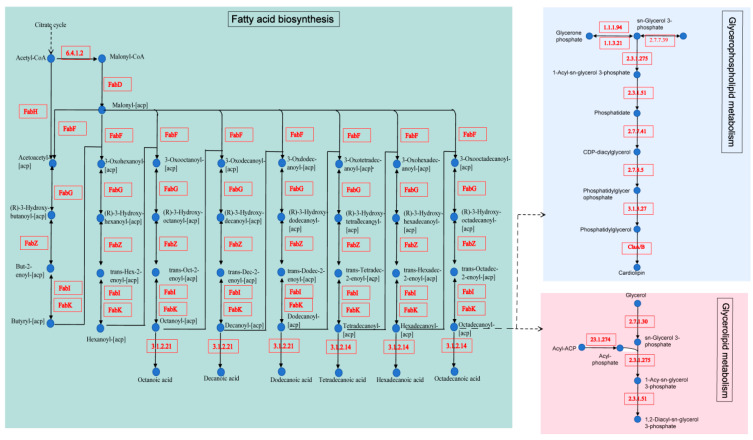
Fatty acid synthesis pathway in *L. plantarum* B22.

**Table 1 foods-14-02354-t001:** Characteristics of the *L. plantarum* B22 genome.

Attributes	Values
Genome Size (bp)	3,290,520
G + C Content (%)	44.57
Coding Gene Number	3148
Coding Gene Average Length (bp)	893.53
rRNA	16
5S rRNA	6
16S rRNA	5
23S rRNA	5
tRNA	69
sRNA	42
Plasmids	1

**Table 2 foods-14-02354-t002:** Information and bioinformatic analysis of potential esterase/lipase in *L. plantarum* B22 genome.

Gene ID	Annotation	Number of Amino Acids	Molecular Weight (Da)	Theoretical pI	Instability Index	Extinction Coefficients	Aliphatic Index	Grand Average of Hydropathicity (GRAVY)	Alpha Helix (Hh, %)	Extended Strand (Ee, %)	Random Coil (Cc, %)
gene0648	Esterase/lipase	248	27,852.40	5.67	36.95	26,930	85.40	−0.215	45.56	16.53	37.90
gene0798	Acetyl esterase/lipase	346	37,837.99	4.85	38.44	41,370	97.34	0.047	43.79	12.13	44.08
gene0817	Acetyl esterase/lipase	229	24,892.17	5.39	26.37	51,910	95.15	−0.022	29.55	21.21	49.24
gene1511	lipase activity	250	28,778.94	8.00	35.86	41,370	93.24	−0.255	42.80	15.20	42.00
gene1598	Lipase_GDSL_2;Lipase_GDSL	314	34,504.56	9.88	21.70	32,890	84.78	−0.326	46.50	12.42	41.08
gene2121	Lipase_GDSL_2;Lipase_GDSL	712	79,929.89	4.80	21.78	83,200	91.01	−0.195	28.99	19.14	51.87
gene2581	Lipase_GDSL_2;Lipase_GDSL	233	25,483.04	8.98	11.94	36,440	97.51	−0.182	43.78	13.73	42.49
gene2576	Monoacylglycerol lipase ABHD6	246	26,816.43	5.78	41.34	21,890	94.84	−0.028	38.62	18.29	43.9
gene2553	Acetyl esterase/lipase	276	30,490.71	6.33	31.11	57,995	90.94	−0.049	27.17	18.84	53.99
gene2578	Acetyl esterase/lipase	469	50,662.08	6.05	21.78	66,935	87.29	−0.113	33.26	14.50	52.24
gene3059	Acetyl esterase/lipase	278	31,534.87	5.45	36.32	55,475	80.04	−0.274	28.06	17.63	54.32
gene3060	Acetyl esterase/lipase	278	30,806.92	5.17	42.24	50,880	86.69	0.014	25.90	19.78	54.32
gene2887	Lipase_GDSL_2;Lipase_GDSL	213	23,106.85	5.12	28.43	32,890	81.64	−0.054	29.58	15.02	55.40

**Table 3 foods-14-02354-t003:** Stress-related genes in *L. plantarum* B22.

Gene ID	Gene Name	Annotation	Gene ID	Gene Name	Annotation
**Universal Stress Family Protein**			**Bile Salt Resistance**		
gene0955	*uspA*	MULTISPECIES: universal stress protein	gene1457	*cfa*	cyclopropane-fatty-acyl-phospholipid synthase family protein
gene1140	*-*	MULTISPECIES: universal stress protein	gene2753	*cfa*	cyclopropane-fatty-acyl-phospholipid synthase family protein
gene1462	*-*	MULTISPECIES: universal stress protein	gene1067	*perM*	AI-2E family transporter
gene1498	*-*	universal stress protein	gene1941	*perM*	MULTISPECIES: AI-2E family transporter
gene2395	*uspA*	MULTISPECIES: universal stress protein	**Oxidative stress**		
gene2317	*uspA*	universal stress protein	gene3072	*katE*	catalase
gene2070	*uspA*	MULTISPECIES: universal stress protein	gene0205	*btuE*	MULTISPECIES: glutathione peroxidase
gene2609	*-*	universal stress protein	gene2956	*yfeX*	Dyp-type peroxidase
gene2523	*uspA*	MULTISPECIES: universal stress protein	gene0068	*arsC*	MULTISPECIES: arsenate reductase (thioredoxin)
gene3149	*-*	MULTISPECIES: universal stress protein	gene0217	*trxA*	MULTISPECIES: thioredoxin
**Acid stress**			gene0617	*trxB*	MULTISPECIES: thioredoxin-disulfide reductase
gene2091	*atpA*	MULTISPECIES: F0F1 ATP synthase subunit alpha	gene1009	*trxA*	MULTISPECIES: thioredoxin family protein
gene2088	*atpC*	MULTISPECIES: F0F1 ATP synthase subunit epsilon	gene2016	*trxA*	MULTISPECIES: thioredoxin
gene2090	*atpG*	MULTISPECIES: F0F1 ATP synthase subunit gamma	gene2308	*-*	MULTISPECIES: thioredoxin family protein
gene2095	*atpB*	F0F1 ATP synthase subunit A	gene2960	*trxA*	MULTISPECIES: thioredoxin family protein
gene2089	*atpD*	MULTISPECIES: F0F1 ATP synthase subunit beta	gene2056	*tpx*	thiol peroxidase
gene2094	*atpE*	MULTISPECIES: F0F1 ATP synthase subunit C	gene0252	*mntH*	manganese transport protein
gene2093	*atpF*	MULTISPECIES: F0F1 ATP synthase subunit B	gene0895	*mntB*	manganese ABC transporter, permease protein
gene0181	*nhaC*	Na(+)/H(+) antiporter NhaC	gene1575	*ppaC*	manganese-dependent inorganic pyrophosphatase
gene2896	*nhaC*	Na+/H+ antiporter NhaC			
**Bacteriocin**			**Organic acid**		
gene0376	*-*	MULTISPECIES: two-peptide bacteriocin plantaricin JK subunit PlnK	gene0469	ldh	MULTISPECIES: L-lactate dehydrogenase
gene0388	*-*	MULTISPECIES: two-peptide bacteriocin plantaricin EF subunit PlnF	gene0716	ldhA	MULTISPECIES: D-2-hydroxyacid dehydrogenase
gene0389	*-*	MULTISPECIES: two-peptide bacteriocin plantaricin EF subunit PlnE			
gene0377	*-*	MULTISPECIES: two-peptide bacteriocin plantaricin JK subunit PlnJ			

**Table 4 foods-14-02354-t004:** Analysis of the antibiotic susceptibility of *L. plantarum* B22.

Antibiotic	Specifications (μg/Piece)	Inhibition Zones Diameter (mm)	Antibiotic Susceptibility	Antibiotic	Specifications (μg/Piece)	Inhibition Zones Diameter (mm)	Antibiotic Susceptibility
Penicillin	1	27.60 ± 2.84	S	Vancomycin	30	0.00 ± 0.00	R
Piperacillin	100	35.27 ± 0.33	S	Chloramphenicol	30	0.00 ± 0.00	R
Ampicillin	10	31.89 ± 3.16	S	Cefalexin	30	0.00 ± 0.00	R
Kanamycin	30	17.28 ± 0.46	I	Cephazolin	30	0.00 ± 0.00	R
Streptomycin	10	0.00 ± 0.00	R	Cefuroxim	30	41.55 ± 0.72	S
Gentamicin	10	18.71 ± 3.12	I	Ceftazidime	30	25.51 ± 2.80	S
Amikacin	10	0.00 ± 0.00	R	Ceftiaxone	30	39.43 ± 1.31	S
Tetracycline	30	17.59 ± 1.10	I	Cefoperazone	30	24.61 ± 0.99	S
Doxycycline	30	0.00 ± 0.00	R	Erythromycin	15	27.42 ± 1.46	S
Minocycline	30	40.09 ± 3.46	S	Lincomycin	2	17.38 ± 1.71	I

Note: Antibiotic susceptibility interpretations (S = Susceptible, I = Intermediate, R = Resistant).

**Table 5 foods-14-02354-t005:** Virulence factor annotation of *L. plantarum* B22.

Gene ID	Location	VFcategory	Related Genes	Gene ID	Location	VFcategory	Related Genes
gene1376	Chromosome	Stress survival	*recN*	gene1391	Chromosome	Immune modulation	*rpe*
gene1574	Chromosome	Stress survival	*msrA*	gene0980	Chromosome	Immune modulation	*rfbD*
gene3072	Chromosome	Stress survival	*katA*	gene0700	Chromosome	Immune modulation	*oatA*
gene0637	Chromosome	Stress survival	*clpP*	gene3134	Chromosome	Immune modulation	*msbA*
gene1086	Chromosome	Stress survival	*clpE*	gene0620	Chromosome	Immune modulation	*manB*
gene0833	Chromosome	Stress survival	*clpC*	gene0428	Chromosome	Immune modulation	*lpxA*
gene3034	Chromosome	Stress survival	*bsh*	gene0990	Chromosome	Immune modulation	*kfiC*
gene1707	Chromosome	Regulation	*relA*	gene0614	Chromosome	Immune modulation	*hasC*
gene0051	Chromosome	Regulation	*phoP*	gene1761	Chromosome	Immune modulation	*gtrB*
gene1321	Chromosome	Regulation	*mprA*	gene1319	Chromosome	Immune modulation	*gndA*
gene2213	Chromosome	Regulation	*bvrR*	gene2993	Chromosome	Immune modulation	*galE*
gene2701	Chromosome	Nutritional/Metabolic factor	*shuU*	gene1633	Chromosome	Immune modulation	*ddrA*
gene1477	Chromosome	Nutritional/Metabolic factor	*pvdH*	gene0321	Chromosome	Immune modulation	*cpsK*
gene2371	Chromosome	Nutritional/Metabolic factor	*purCD*	gene0966	Chromosome	Immune modulation	*cpsI*
gene1281	Chromosome	Nutritional/Metabolic factor	*narH*	gene0664	Chromosome	Immune modulation	*cpsG*
gene2930	Chromosome	Nutritional/Metabolic factor	*mgtB*	gene1743	Chromosome	Immune modulation	*cpsB*
gene2648	Chromosome	Nutritional/Metabolic factor	*iroC*	gene0996	Chromosome	Immune modulation	*cps4I*
gene2783	Chromosome	Nutritional/Metabolic factor	*fbpC*	gene1561	Chromosome	Immune modulation	*bcs1′*
gene1718	Chromosome	Nutritional/Metabolic factor	*dhbF*	gene2003	Chromosome	Biofilm	*bopD*
gene2358	Chromosome	Nutritional/Metabolic factor	*carB*	gene1864	Chromosome	Adherence	*tufA*
gene2359	Chromosome	Nutritional/Metabolic factor	*carA*	gene1542	Chromosome	Adherence	*pavA*
gene1434	Chromosome	Motility	*flmH*	gene3148	Chromosome	Adherence	*lap*
gene2249	Chromosome	Immune modulation	*wbuZ*	gene2078	Chromosome	Adherence	*IlpA*
gene0976	Chromosome	Immune modulation	*wbtL*	gene0590	Chromosome	Adherence	*groEL*
gene0802	Chromosome	Immune modulation	*wbtH*				

## Data Availability

The original contributions presented in the study are included in the article/Appendix A, further inquiries can be directed to the corresponding author.

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
