# Peer review of "Dissecting the High Esterase/Lipase Activity and Probiotic Traits in Lactiplantibacillus plantarum B22: A Genome-Guided Functional Characterization"

_foods, 2025, doi:10.3390/foods14132354_

Round 1

Reviewer 1 Report

Comments and Suggestions for Authors

In this study by Chai et al. (2025), the authors investigated the probiotic and enzymatic potential of Lactobacillus plantarum from a molecular and phenotypic perspective. The work is interesting and quite complete. Some points need to be clarified before the manuscript is published. The English also needs to be improved, and perhaps the issues are a result of this.

Major comments

  1. line 636-643: Although the authors emphasize the safety of using this strain. The context of the variables may allow for different interpretations. What about other strains with high esterase and lipase capacity such as S. aureus and Mycobacteria? Actually, I missed comparing how this enzymatic activity of the probiotic differs from pathogenic strains. What prevents L. plantarum from being potentially pathogenic, specifically regarding these metabolisms?
  2. Line 320 and Figure 1E: this isn't electron microscopy, its optical microscopy, please correct.
  3. Considering the ISAPP definition of probiotics and their strain-specific nature (see “The International Scientific Association for Probiotics and Prebiotics consensus statement on the scope and appropriate use of the term probiotic”), it is unclear how a wild bacterium isolated from food can be regarded as the same microorganism—specifically L. plantarum B22. While molecular similarities may confirm they belong to the same species, this does not necessarily imply relevance at the strain or lineage level. In my view, the isolates analyzed in this study are distinct and should be assigned their own identification or code. In fact, if the isolate is indeed so similar that it can be considered B22, then this study—and its probiotic characterization—would not be justified, as the original strain’s properties have already been established by its initial isolator in order to qualify it as a probiotic. Please clarify how this is possible.

Minor comments

  1. General: The figures and tables are not formatted properly, they are distorted, tables with fragmented titles, please review.
  2. General: Please check that all figure legends are appropriate and allow for independent interpretation. Also check that they are presented in the correct order throughout the text.
  3. Line 114: at what temperature?
  4. line 123-124: reference the study from which the strains were obtained.
  5. Line 126: what is the composition of this medium?
  6. General: what was the final pH of each culture medium used?
  7. line 142-143: please clarify.
  8. line 148-149: specify the concentrations of each reagent.
  9. line 152: what is the spin condition?
  10. line 155-156: specify all the experimental conditions.
  11. line 233; 244: it is not clear whether the cell suspension or the supernatant of the microorganisms was evaluated, please specify.
  12. Line 270: add reference
  13. Line 278 or Table 4: include the concentration of each of the disks containing antimicrobials.
  14. line 519; 526; 634 and where it is necessary: check formatting.
  15. line 589: I don't remember the description of these analyses in the methodology.
  16. Lines 25-26: are contradicted in lines 613-615, with regard to antimicrobials. Remove or modify the abstract.

Comments on the Quality of English Language

Please review the sentence structures for greater fluidity and cohesion.

Author Response

Comments and Suggestions for Authors

In this study by Chai et al. (2025), the authors investigated the probiotic and enzymatic potential of Lactobacillus plantarum from a molecular and phenotypic perspective. The work is interesting and quite complete. Some points need to be clarified before the manuscript is published. The English also needs to be improved, and perhaps the issues are a result of this.

Dear Editors and Reviewers:

We appreciate editor and reviewers very much for positive and constructive comments and suggestions on our manuscript entitled "Dissecting the High Esterase/Lipase Activity and Probiotic Traits in Lactiplantibacillus plantarumB22: A Genome-Guided Functional Characterization".

We appreciate the reviewers' comments and have carefully addressed each point raised. Below, we provide a detailed response to each comment and describe the modifications made to the manuscript. Additionally, we have used a different colored text to highlight the changes within the revised manuscript. These changes have improved the manuscript considerably and we hope that it can be interesting manuscript for journal readers.

Thank you once again for your assistance and consideration. We look forward to the final decision.

Looking forward to hearing from you.

Sincerely,

Best Regards,

Aixiang Huang

aixianghuang@126.com

Point to point response to reviewer’s comments are mentioned below:

Major comments

  1. line 636-643: Although the authors emphasize the safety of using this strain. The context of the variables may allow for different interpretations. What about other strains with high esterase and lipase capacity such as S. aureus and Mycobacteria? Actually, I missed comparing how this enzymatic activity of the probiotic differs from pathogenic strains. What prevents plantarumfrom being potentially pathogenic, specifically regarding these metabolisms?

The authors’ answer: Thank you for the reviewer’s valuable feedback. We greatly appreciate your expertise and the opportunity to clarify these important points. Your comments have significantly strengthened the manuscript.

(1)We added comparisons with Staphylococcus aureus and Bacillus divergens reported to have high lipase activity.

Lines 726-732: Staphylococcus aureus and Mycobacterium species have high esterase and lipase activities and utilize these enzymes for invasive and virulence effects such as disrupting host cell membranes and promoting tissue invasion [1]. In contrast, L. plantarum, while possessing esterase/lipase capabilities, primarily employs these enzymes to metabolize dietary lipids and generate flavor compounds [2]. Furthermore, L. plantarum has an extensive history of use, lacks genes for hemolytic toxins, and does not elicit adverse reactions in the host [3].

(2)Factors that prevent Lactobacillus plantarum from exhibiting potential pathogenicity in these metabolic processes are:

  • Lactic acid bacteria (LAB) are used extensively in foods because they are generally recognized as safe (GRAS) organisms [4]. In addition, Lactobacillus plantarum is one of the strains in China's list of Strains that can be used in food [5]. Lactobacillus plantaruma widespread lactic acid bacteria commonly found many fermented food products and gastrointestinal tract, which employed as probiotics [6]. It has not been found to be pathogenic.
  • Functional Role and Target Substrates: In contrast, in plantarumand other LAB, esterases and lipases primarily function in food fermentation and environmental adaptation. They hydrolyze dietary fats and esters present in the food matrix (e.g., milk fats, plant oils) or in the intestinal lumen, contributing to flavor development (ester formation/lipolysis) [7]. Their activity is directed towards dietary or environmental substrates, not host tissues or cellular components.
  • Absence of Key Virulence Machinery: Lactobacillus plantarum, although possessing metabolic enzymes such as esterases/lipases, is completely lacking in the core virulence factors required by pathogens such as aureusand Mycobacterium:e.g., it does not produce toxins such as hemolysins [8], as demonstrated by the hemolysis experiments in the manuscript (Lines636-653). It lacks a mechanism to actively invade host cells. It lacks the complex mechanisms by which pathogenic bacteria actively suppress or destroy immunity in a deleterious manner (e.g., superantigens, ESX secretion system) [9].

We hope this detailed explanation and the revised manuscript text effectively address your valid concerns by providing the explicit comparison you requested and clearly articulating the fundamental biological reasons why metabolic capabilities like esterase/lipase activity in L. plantarum are not associated with pathogenicity. Thank you again for your invaluable contribution to improving our manuscript.

References support:

  • Nguyen, M. T., Luqman, A., Bitschar, K., Hertlein, T., Dick, J., Ohlsen, K., ... & Götz, F. (2018). Staphylococcal (phospho) lipases promote biofilm formation and host cell invasion. International Journal of Medical Microbiology, 308(6), 653-663.https://doi.org/10.1016/j.ijmm.2017.11.013
  • Xia, L., Qian, M., Cheng, F., Wang, Y., Han, J., Xu, Y., ... & **, Y. (2023). The effect of lactic acid bacteria on lipid metabolism and flavor of fermented sausages. Food Bioscience, 56, 103172.https://doi.org/10.1016/j.fbio.2023.103172
  • Nieuwboer, M. V. D. ,  Hemert, S. V. ,  Claassen, E. , &  Vos, W. M. D. . (2016). Lactobacillus plantarumwcfs1 and its host interaction: a dozen years after the genome. Microbial Biotechnology, 9(4). https://doi.org/1111/1751-7915.12368
  • Baugher JL,Durmaz E,Klaenhammer TR. Spontaneously induced prophages in Lactobacillus gasseri contribute to horizontal gene transfer. Appl Environ Microbiol. 2014;80 (11):3508-17. https://doi.org/1128/AEM.04092-13
  • https://www.nhc.gov.cn/cms-search/downFiles/f7177b32d93142f9827dcc55fb6889c2.pdf.
  • José María Landete,José Antonio Curiel,Héctor Rodríguez, et al. Aryl glycosidases from Lactobacillus plantarum increase antioxidant activity of phenolic compounds. Journal of Functional Foods. 2014;7 (0):322-329. https://doi.org/1016/j.jff.2014.01.028
  • Zhao, W., Cao, Q., Shi, Z., Chen, C., Zhou, C., Sun, Y., ... & Fan, X. (2025). Insights into the molecular mechanisms of microbial agent fermentation on the formation of sensory quality in air-dried geese via metabolomics. LWT, 215, 117218.https://doi.org/10.1016/j.lwt.2024.117218
  • Grujović, M. Ž., Marković, K. G., Morais, S., & Semedo-Lemsaddek, T. (2024). Unveiling the Potential of Lactic Acid Bacteria from Serbian Goat Cheese. Foods, 13(13), 2065.https://doi.org/10.3390/foods13132065
  • Conrad, W. H., Osman, M. M., Shanahan, J. K., Chu, F., Takaki, K. K., Cameron, J., ... & Ramakrishnan, L. (2017). Mycobacterial ESX-1 secretion system mediates host cell lysis through bacterium contact-dependent gross membrane disruptions. Proceedings of the National Academy of Sciences, 114(6), 1371-1376.https://doi.org/10.1073/pnas.1620133114
  1. Line 320 and Figure 1E: this isn't electron microscopy, its optical microscopy, please correct.

The authors’ answer: Thank you for pointing out the error. We have corrected the description in line 356 and the caption of Figure 1E (line 376) from "electron microscopy" to "optical microscopy".

  1. Considering the ISAPP definition of probiotics and their strain-specific nature (see “The International Scientific Association for Probiotics and Prebiotics consensus statement on the scope and appropriate use of the term probiotic”), it is unclear how a wild bacterium isolated from food can be regarded as the same microorganism—specifically plantarumB22. While molecular similarities may confirm they belong to the same species, this does not necessarily imply relevance at the strain or lineage level. In my view, the isolates analyzed in this study are distinct and should be assigned their own identification or code. In fact, if the isolate is indeed so similar that it can be considered B22, then this study—and its probiotic characterization—would not be justified, as the original strain’s properties have already been established by its initial isolator in order to qualify it as a probiotic. Please clarify how this is possible.

The authors’ answer: Thank you for raising this important point. In our study, the strain involved was isolated from food and molecularly identified as having 99% similarity to Lactiplantibacillus plantarum. Based on this high molecular similarity, we classified the strain as Lactiplantibacillus plantarum and assigned it the number B22 to facilitate its differentiation and identification in this study. This identification aligns with the findings of Kumar et al. [1], who effectively identified unknown strains through a combination of morphological features and molecular methods, demonstrating the reliability of this approach for LAB species identification. Similarly, Jiang et al. [2] utilized morphological observations and phylogenetic tree analysis to successfully identify a strain as Lactobacillus salivarius.

References support:

  • Kumar, A.,Joishy, T.,Das, S.,Kalita, M.C,Mukherjee, A.K, & Khan, M.R (2022). A Potential Probiotic Lactobacillus plantarum JBC5 Improves Longevity and Healthy Aging by Modulating Antioxidative, Innate Immunity and Serotonin-Signaling Pathways in Caenorhabditis elegans. Antioxidants (Basel, Switzerland), 11 (2). https://doi.org/10.3390/antiox11020268
  • Jiang, Y. H., Yang, R. S., Lin, Y. C., Xin, W. G., Zhou, H. Y., Wang, F., ... & Lin, L. B. (2023). Assessment of the safety and probiotic characteristics of Lactobacillus salivarius CGMCC20700 based on whole-genome sequencing and phenotypic analysis. Frontiers in Microbiology, 14, 1120263.  https://doi.org/10.3389/fmicb.2023.1120263

Minor comments

4.General: The figures and tables are not formatted properly, they are distorted, tables with fragmented titles, please review.

The authors’ answer: Thank you for your feedback. We have carefully reviewed and reformatted all figures and tables to ensure they are properly formatted and not distorted. We have also ensured that all table titles are complete and non-fragmented. (line 372: Figure 1; line 418: Figure 2; line 464: Figure 3; line 470: Table 2; line 586: Figure 4; line 595: Table 3; line 702: Table 4; line 734: Table 5;). In addition, we have attached revised figures and tables to the “Figures, Graphics, Images-re” file and the “Supplementary File-re” file.

5.General: Please check that all figure legends are appropriate and allow for independent interpretation. Also check that they are presented in the correct order throughout the text.

The authors’ answer: Dear reviewer, thank you for your suggestion. We have carefully checked and revised all figure legends to ensure they are appropriate. We have also verified that they are presented in the correct order throughout the text. (lline 372: Figure 1; line 418: Figure 2; line 464: Figure 3; line 470: Table 2; line 586: Figure 4; line 595: Table 3; line 702: Table 4; line 736: Table 5; lines 1189-1208:Table S1-S2, Figure S1-S4).

6.Line 114: at what temperature?

The authors’ answer: Thank you for pointing out the missing information. We have revised the sentence at Lines 121-122 and added the fermentation temperature.

Lines 124-126: LAB was cultured in MRS broth (pH 6.5) at 37°C for 24 h with an inoculation concentration of 1% (v/v)..

7.line 123-124: reference the study from which the strains were obtained.

The authors’ answer: Thank you for your valuable suggestion. We have updated the manuscript by adding Table S1, which clearly indicates that these LAB strains were isolated from Yunnan specialty resources by our laboratory (line 122 and line 1188: Table S1). This table provides detailed information about the origin of the strains to address your concern. We believe this addition strengthens the clarity and completeness of the study.”

8.Line 126: what is the composition of this medium?

The authors’ answer: Thank you for the reviewer’s valuable feedback. The neutral red medium employed for the preliminary screening of esterase/lipase producers consists of the following components:

Lines 135-139: For preliminary screening, 20 μL of the fermentation broth of LAB cultured to the logarithmic phase was spread onto neutral red oil decomposition agar plates (composed of 10 g/L peptone, 5 g/L beef extract, 5 g/L NaCl, 10 g/L olive oil, 17 g/L agar, and 1 mL/L of 1.6% (w/v) neutral red aqueous solution, final pH 7.2), which were then incubated at 37°C for 24 h.

9.General: what was the final pH of each culture medium used?

The authors’ answer: Thank you for your attention to detail. We have supplemented the final pH of each culture medium used in the study. The details are as follows:

Lines 124-126: LAB was cultured in MRS broth (pH 6.5) at 37°C for 24 h with an inoculation concentration of 1% (v/v). 

Lines 141-143: A 0.8 cm hole was punched in a glycerol tributyrate agar (pH 7.0) plate containing tryptone (2.5 g/L), casein peptone (2.5 g/L), yeast extract (3 g/L), NaCl (5 g/L), glycerol tributyrate (1%, v/v), and agar (17 g/L).

Lines 242-244: The bile salt tolerance of the strains was evaluated by adding 1.0 mL of the L. plantarum B22 bacterial suspension to MRS medium (9 mL, pH 6.5) containing 0.3% (w/v) bile.

Lines 289-291: The indicator bacteria were inoculated into Luria Bertani (LB) medium (pH 7.2) with an inoculum volume fraction of 1%.

Lines 299-301: Following a modified version of Lu et al. [40]. protocol, activated L. plantarum B22 was inoculated onto Columbia blood agar plates (Oxoid, Thermo Fisher Scientific, UK, pH 7.3) and incubated at 37°C for 48 h.

Lines 306-309: The activated L. plantarum B22 was inoculated in the amino acid decarboxylase induction medium containing arginine, lysine, and ornithine decarboxylase biochemical tubes (pH 6.0), respectively, and cultured at 37°C for 24 h.

Lines 314-316: The antibiotic drug sensitivity test was performed according to the disk diffusion method, where a 108 CFU/mL culture of L. plantarum B22 (200 µL) was spread evenly onto MRS agar (pH 6.5).

10.line 142-143: please clarify.

The authors’ answer: Thank you for your attention to detail. We have clarified the determination of enzyme activity in the text.

Lines 157-163: One unit of enzyme activity (U) was defined as the amount of enzyme that releases 1 μmol of p-nitrophenol per minute under specified conditions [1].

Esterases/Lipases activity (U/mL)=cV/tV'

Where c is the concentration of p-nitrophenol (μmol/L); V is the final volume of the reaction solution after acid-base adjustment (mL); V' is the volume of the enzyme solution used (mL); t is the incubation time (min).

11.line 148-149: specify the concentrations of each reagent.

The authors’ answer: Thank you very much for your suggestion. We have now specified the concentrations of each reagent in lines 148-149.

Lines 167-169: After centrifugation, the pellet was ground in liquid nitrogen and lysed with 0.5% SDS lysis buffer, containing proteinase K (0.4 mg/ml) and 0.5% mercaptoethanol.

12.line 152: what is the spin condition?

The authors’ answer: Thank you very much for the review’s suggestion. We have now added the information regarding the spin condition in the manuscript.

Lines 171-172: DNA was then precipitated by the addition of isopropanol, followed by gentle mixing and centrifugation at 8000 g for 10 min at 4℃.

13.line 155-156: specify all the experimental conditions.

The authors’ answer: Thank you for your suggestion. We have revised sentences in the manuscript. The revised sentence is as follows:

Lines 174-180: Finally, the DNA was assessed for quality using a NanoDrop spectrophotometer (NanoDrop Technologies, Inc., Wilmington, DE) at a wavelength of 260 nm and 280 nm to measure the absorbance and calculate the A260/A280 ratio, and using a Qubit fluorometer (Life Technologies, Inc., Carlsbad, CA, USA) with the Qubit dsDNA BR Assay Kit (Fisher Scientific, Waltham, MA, USA)to determine the concentration.

14.line 233; 244: it is not clear whether the cell suspension or the supernatant of the microorganisms was evaluated, please specify.

The authors’ answer: Thank you for your insightful comment. We have now revised the text to clarify this point. 

Lines 264-266: In this assay, 2 mL of a 0.4 mM DPPH solution in methanol was mixed with either 2 mL of distilled water (as the control) or the L. plantarum B22 CFS.

Lines 277-279: In the assay, 150 μL of either the L. plantarum B22 CFS or distilled water (control) was mixed with 150 μL of the ABTS solution and incubated at 37°C for 10 min.

15.Line 270: add reference

The authors’ answer: Thank you for your suggestion. We have now added the relevant reference to support the method used in the assay.

Lines 305-306: The amino acid decarboxylase test according to the method described by Li et al. [41] with some modifications.

16.Line 278 or Table 4: include the concentration of each of the disks containing antimicrobials.

The authors’ answer: Thank you for your suggestion. We have included the concentration of each of the disks containing antimicrobials in Table 4 (line 701).

17.line 519; 526; 634 and where it is necessary: check formatting.

The authors’ answer: Thank you for your suggestion. We have carefully checked the formatting of lines 575, 582, 712 and other relevant sections. All identified formatting issues have been corrected in the revised. 

18.line 589: I don't remember the description of these analyses in the methodology.

The authors’ answer: Thank you for raising this important point. The relevant methodology is detailed in section "2.6.1. Detection of Harmful Metabolites of L. plantarum B22" under "Amino Acid Decarboxylase Test". We apologize for the confusion caused by the missing subheading number, which may have led to difficulties in locating this information. We have now revised the paper to include the correct subheading number, ensuring that the description is easily identifiable.

Line 304-312: (2) Amino Acid Decarboxylase Test

The amino acid decarboxylase test according to the method described by Li et al. [41] with some modifications.The activated L. plantarum B22 was inoculated in the amino acid decarboxylase induction medium containing arginine, lysine, and ornithine decarboxylase biochemical tubes (pH 6.0), respectively, and cultured at 37°C for 24 h. Biochemical tubes without L. plantarum B22 inoculation served as controls. The tubes were sealed with liquid paraffin and incubated at 37°C for 24 h. A color change in the test tube to purple indicates a positive result, while yellow indicates a negative result.

19.Lines 25-26: are contradicted in lines 613-615, with regard to antimicrobials. Remove or modify the abstract.

The authors’ answer: Thank you for the reviewer’s valuable feedback. We have carefully reviewed the content and have modified the abstract to remove the inconsistencies related to the antimicrobials (lines 31-33).

Comments on the Quality of English Language

Please review the sentence structures for greater fluidity and cohesion.

The authors’ answer: Dear reviewer, Thank you for your feedback. We will further edit the English language to ensure that the ideas are communicated effectively. The manuscript is checked for English editing by Dr. Adhita Sri Prabakusuma who is responsible for all the English editing of our manuscripts. His google scholar account is as follows:

(https://www.researchgate.net/profile/Adhita-Sri-Prabakusuma).

In addition, we greatly appreciate the guidance from the reviewers on language, which has significantly helped us improve the manuscript.

Reviewer 2 Report

Comments and Suggestions for Authors

Dear authors/editor,

This manuscript “Dissecting the High Esterase/Lipase Activity and Probiotic Traits in Lactiplantibacillus plantarum B22: A Genome-Guided Functional Characterization” investigates esterase/lipase activity and probiotic traits in the microorganism Lactiplantibacillus plantarum B22 through genome-guided functional characterization.

The study is timely and relevant. It provides meaningful insights into the importance of assessing esterase/lipase activity in probiotic foods and demonstrates through various toxicity tests that the probiotic involving the use of this microorganism is safe and harmless for human consumption, a globally important issue. However, several substantial issues must be addressed before the paper can be considered for publication.

Abstract

The abstract should comprise a concise exposition of the research problem, the methodology employed, the most salient results, and the primary conclusions. However, in this case, the study commences abruptly, eschewing any introductory remarks or methodological elaboration. It is recommended that this section be restructured.

Abbreviations should only be used in this section and subsequent sections of the manuscript after the meaning of the abbreviated term has been indicated at least once (e.g. GC line 16).

Introduction

The final section of the introduction must be revised to ensure that it clearly and concisely articulates the objectives of the research.

Materials and Methods

This section of the document is well written, organized and structured. However, some references to methodological procedures have been omitted and need to be added, particularly in the concluding paragraphs. Lines: 132, 184-195, 143, 156, 166, 179, 185, 203, 212, 216, 228, 276.

The units of measurement used for the different variables/magnitudes assessed should conform to the International System of Units (SI). For example, 'microlitres' is incorrectly used in line 280; the correct term is 'µL'.

Some minor spelling errors need to be addressed. For example, in lines 289 and 291, the correct words are 'deviation' and 'considered'.

Clarity of Statistical Methodology: While ANOVA is appropriately applied, no justification or assumptions are discussed (e.g., normality, homogeneity of variance). This weakens the reliability of significance testing.

Results and Discussion

The Results and Discussion section presents the research findings and their interpretation in a clear and concise manner, discussing them in the context of previous knowledge and answering the research questions posed. However, I would like to make a couple of suggestions:

  • It is recommended that Figures 1A and 1B be enlarged to enhance their resolution, as they are not sufficiently visible in their current presentation.
  • Scientific names of species are written according to binomial nomenclature. The first word is the genus and is capitalised, while the second word is the specific name or epithet and is written in lowercase. Both are written in italics. Therefore, the name Candida rugosa (line 394) must conform to this international guideline.
  • It is recommended that Table 2 be displayed on a landscape page, thus ensuring that the content is not excessively cluttered.
  • It is imperative that discrepancies observed between disparate treatments or experiments are meticulously documented and deliberated, with a view to identifying potential causative factors.
  • It is imperative that the term 'Glycerophospholipid metabolism' is highlighted in bold in Table S1, to guarantee uniformity of content.
  • As illustrated in Table 4, it is imperative to specify the definitions of the various antibiotic susceptibilities, namely S, R and I.
  • Although the study is very comprehensive and provides a thorough discussion of the topic, it lacks an identification of the weaknesses and limitations of the research conducted, as well as any prospects for future research. I therefore suggest mentioning these at the end of the Results and Discussion section.

Conclusions

The research conclusions are a concise and clear summary of the results obtained, addressing the initial objectives and answering the formulated research questions. I therefore consider them to be very appropriate and in line with the work carried out.

The manuscript is of high quality and makes a significant contribution to the field of probiotic food safety assessment by using genomic tools. However, issues of methodological transparency must be addressed thoroughly. Attention must also be paid to the statistical procedures applied, the labelling of figures and how they are presented.

Taking the above into account, I believe that the manuscript requires only minor revisions.

I look forward to hearing from you in response to my concerns.

Author Response

Comments and Suggestions for Authors

Dear authors/editor,

This manuscript “Dissecting the High Esterase/Lipase Activity and Probiotic Traits inLactiplantibacillus plantarumB22: A Genome-Guided Functional Characterization” investigates esterase/lipase activity and probiotic traits in the microorganismLactiplantibacillus plantarumB22 through genome-guided functional characterization.

The study is timely and relevant. It provides meaningful insights into the importance of assessing esterase/lipase activity in probiotic foods and demonstrates through various toxicity tests that the probiotic involving the use of this microorganism is safe and harmless for human consumption, a globally important issue. However, several substantial issues must be addressed before the paper can be considered for publication.

Dear Editors and Reviewers:

Thank you for your constructive comments and suggestions for our manuscript “Dissecting the High Esterase/Lipase Activity and Probiotic Traits in Lactiplantibacillus plantarum B22: A Genome-Guided Functional Characterization”. We are grateful for your recognition of the timeliness and relevance of our study, as well as the insights it provides into the significance of evaluating esterase/lipase activity in probiotic foods.

We appreciate the reviewers' comments and have carefully addressed each point raised. Below, we provide a detailed response to each comment and describe the modifications made to the manuscript. Additionally, we have used a different colored text to highlight the changes within the revised manuscript. These changes have improved the manuscript considerably and we hope that it can be interesting manuscript for journal readers.

Thank you again for your time and valuable input. We look forward to the final decision.

Looking forward to hearing from you.

Sincerely,

Best Regards,

Aixiang Huang

aixianghuang@126.com

Point to point response to reviewer’s comments are mentioned below:

Reviewers' comments: 

1.Abstract

The abstract should comprise a concise exposition of the research problem, the methodology employed, the most salient results, and the primary conclusions. However, in this case, the study commences abruptly, eschewing any introductory remarks or methodological elaboration. It is recommended that this section be restructured.

The authors’ answer: Dear reviewer, Thank you for your valuable feedback. We have restructured the abstract to include a concise introduction to the research problem, an overview of the methodology, the key results, and the primary conclusions

Lines 13-36: ABSTRACT

Lactiplantibacillus plantarum B22 exhibits high esterase/lipase activity, but the genomic and probiotic potential remains unclear. We employed an integrated approach combining whole-genome sequencing, molecular docking studies, and phenotypic assays to dissect the genomic and functional basis underlying the high lipolytic activity and probiotic traits of L.plantarum B22. This strain exhibited robust lipase activity (3.45 ± 0.13 U/mL), with whole-genome analysis revealed that the complete genome of this strain spans 2,027,325 bp, encoding 2,005 genes with a guanine-cytosine (GC) content of 35.06%. Notably, 13 esterase/lipase genes were identified, four of which (gene3060, gene3059, gene2553, gene0798) harbor conserved catalytic triads (Ser-His-Gly/Ala) essential for lipase function. Molecular docking studies confirmed strong binding affinity to tributyrin (ΔG ≤ –5.52 kcal/mol) and elucidated the interaction mechanisms, involving hydrogen bonding and hydrophobic interactions between the esterase/lipase enzymes and tributyrin. Phenotypic and genomic analyses further demonstrated that L. plantarum B22 possesses excellent tolerance to simulated human gastrointestinal tract conditions, along with potent antioxidant and antimicrobial activities, highlighting its strong probiotic potential. Genomic annotation also identified 68 genes assoicated with lipid metabolism and an intact fatty acid synthesis pathway. Importantly, the analysis of phenotypes and genes involved in virulence factors, and the production of harmful metabolites suggests that L. plantarum B22 is safe. Collectively, this study offers novel insights into the genome-guided functional characterization of L. plantarum B22, providing a robust foundation for its development as a functional probiotic strain.

  1. Abbreviations should only be used in this section and subsequent sections of the manuscript after the meaning of the abbreviated term has been indicated at least once (e.g. GC line 16).

The authors’ answer: Thank you for your suggestion. We have supplemented the full form of the abbreviation when it first appears in the manuscript.

Line 21: GC content→ guanine-cytosine (GC) content

3.Introduction

The final section of the introduction must be revised to ensure that it clearly and concisely articulates the objectives of the research.

The authors’ answer: Thank you for your suggestion. We have revised the final section of the introduction.

Lines 99-112: The LABs exhibiting lipolytic activity hold significant potential in food biotechnology. In this study, L. plantarum B22, isolated from goat's milk and demonstrating remarkably high esterases/lipases activity Crucially, the genetic determinants (esterase/lipase-encoding genes), enzyme structural features, enzyme substrate binding mechanisms, beneficial probiotic properties, and safety profile of this strain remained unexplored. To address this gap, we employed integrated whole-genome sequencing and phenotypic analyses to systematically investigate the relationship between key phenotypes of L. plantarum B22 and its underlying genomic features. The specific objectives of this study were: (1)To identify and characterize the genetic determinants and structural features responsible for the high lipolytic activity. (2) To elucidate the substrate binding mechanisms of key esterases/lipases using molecular docking. (3) To comprehensively assess the probiotic traits and safety profile. This study provides genomic insights for predicting the function of target strains.

4.Materials and Methods

This section of the document is well written, organized and structured. However, some references to methodological procedures have been omitted and need to be added, particularly in the concluding paragraphs. Lines: 132, 184-195, 143, 156, 166, 179, 185, 203, 212, 216, 228, 276.

The authors’ answer: Thank you for your valuable suggestions! We have carefully reviewed the content and have modified the manuscript.

Lines 145-147: The plates were incubated at 37°C for 24 hours using a caliper (Deli Co., Ningbo, Zhejiang, China) to measure the size of the clear zone (mm) [29].

Lines 159-163: One unit of enzyme activity (U) was defined as the amount of enzyme that releases 1 μmol of p-nitrophenol per minute under specified conditions [1].

Esterases/Lipases activity (U/mL)=cV/tV'

Where c is the concentration of p-nitrophenol (μmol/L); V is the final volume of the reaction solution after acid-base adjustment (mL); V' is the volume of the enzyme solution used (mL); t is the incubation time (min).

Lines 174-180: Finally, the DNA was assessed for quality using a NanoDrop spectrophotometer (NanoDrop Technologies, Inc., Wilmington, DE) at a wavelength of 260 nm and 280 nm to measure the absorbance and calculate the A260/A280 ratio, and using a Qubit fluorometer (Life Technologies, Inc., Carlsbad, CA, USA) with the Qubit dsDNA BR Assay Kit (Fisher Scientific, Waltham, MA, USA) to determine the concentration [30].

Lines 200-204: Genes associated with antibiotic resistance in the L. plantarum B22 genome were predicted using the CARD (Comprehensive Antibiotic Resistance Database). The criteria used were E-value < 1 × 10-2, coverage >70%, and similarity >30%. [31]

Lines 207-209: The genes with function of esterases/lipase were screened through comparing parameters with functional annotations, such as Identity, E_value, and Score. [32]

Lines 208-210: The experiment was performed in accordance with previously reported methods with slight modifications [33,34]. 

Lines 239-242: The viable bacteria were quantified by the plate counting method at 0 h and 3 h, and the survival rate was calculated. [35]

The bile salt tolerance of L. plantarum B22 was determined according to the method reported by Wei et al [36].

Lines 248-249: The tolerance of L. plantarum B22 to simulated gastrointestinal conditions was assessed according to the method reported by Fei et al. [37] with minor modifications. 

Lines 261-262: The L. plantarum B22 cell-free supernatant (CFS) was prepared according to the method described by Wang et al. [38] with some modifications. 

5.The units of measurement used for the different variables/magnitudes assessed should conform to the International System of Units (SI). For example, 'microlitres' is incorrectly used in line 280; the correct term is 'µL'.

The authors’ answer: Thank you for your suggestion, we are very sorry for our incorrect writing. We have revised the manuscript accordingly, and the term in line 315 has been changed from "ul" to "µL".

6.Some minor spelling errors need to be addressed. For example, in lines 289 and 291, the correct words are 'deviation' and 'considered'.

The authors’ answer: Thank you very much for your reminder, we are very sorry for our incorrect writing. We have corrected typo.

Line 324:“devi­ ation” revised to “deviation”.

Line 327:“consid­ ered” revised to “considered”.

7.Clarity of Statistical Methodology: While ANOVA is appropriately applied, no justification or assumptions are discussed (e.g., normality, homogeneity of variance). This weakens the reliability of significance testing.The authors’ answer: Thank you for your valuable feedback. We have revised the statistical analysis section.

Lines 326-328: Normality of data distribution was assessed using the Shapiro–Wilk test and for the homogeneity of variance with Levene’s test. Results were considered statistically significant whenever p-value < 0.05.

8.Results and Discussion

The Results and Discussion section presents the research findings and their interpretation in a clear and concise manner, discussing them in the context of previous knowledge and answering the research questions posed. However, I would like to make a couple of suggestions:

It is recommended that Figures 1A and 1B be enlarged to enhance their resolution, as they are not sufficiently visible in their current presentation.

The authors’ answer: Thank you for your suggestion. We have enlarged Figures 1A and 1B to enhance their resolution, ensuring they are clearly visible and properly presented. (line 372)

  1. Scientific names of species are written according to binomial nomenclature. The first word is the genus and is capitalised, while the second word is the specific name or epithet and is written in lowercase. Both are written in italics. Therefore, the name Candida rugosa (line 394) must conform to this international guideline.

The authors’ answer: Thank you for your valuable feedback. Our opinion modifies Candida Rugosa to Candida rugosa. (line 444)

10.It is recommended that Table 2 be displayed on a landscape page, thus ensuring that the content is not excessively cluttered.

The authors’ answer: Thank you for your suggestion. We have displayed Table 2 on a landscape page, thus ensuring that the content is not too cluttered. (line 470)

11.It is imperative that discrepancies observed between disparate treatments or experiments are meticulously documented and deliberated, with a view to identifying potential causative factors.

The authors’ answer: Thank you for your valuable feedback. We have meticulously documented and discussed the discrepancies observed between disparate treatments or experiments in the manuscript:

Lines 568-573: As shown in Fig 4F, L. plantarum B22 demonstrated significant inhibitory effects against Pseudomonas aeruginosa, Escherichia coli, Salmonella enterica, and Staphylococcus aureus, with varying degrees of effectiveness. Among these, the highest inhibition was observed against S. aureus, with an inhibition zone of 17.70 ± 0.42 mm. This was followed by Salmonella enterica with an inhibition zone of 16.04 ± 1.08 mm. Previous studies have also reported similar findings.

Lines 648-656: In this study, the hemolytic activity of L. plantarum B22 was tested on columbia blood agar plates, with S. aureus ATCC 25923 (a known β-hemolytic strain) used as a positive control. As shown in Fig. S4A-4B, no clear zone of lysis was observed around L. plantarum B22, unlike the S. aureus, indicating that the strain does not produce β-hemolysin. This finding demonstrates that L. plantarum B22 lacks hemolytic pathogenic potential. Moreover, genomic analysis revealed no genes associated with hemolysis in the L. plantarum B22 genome, further confirming the absence of hemolytic activity at the molecular level.

Lines 665-672: As shown in the Fig. S4C-4D L. plantarum B22 tested negative for amino acid decarboxylase activity. Crucially, whole-genome sequencing revealed no intact genes encoding tyrosine (tdc), arginine (adc), histidine (hdc), or lysine (ldc) decarboxylases, and no complete gene clusters for biosynthesizing tyramine, putrescine, histamine, or cadaverine. This genotype-phenotype concordance definitively establishes that L. plantarum B22 does not produce amino acid decarboxylases and, therefore, does not decompose amino acids to form biogenic amines.

12.It is imperative that the term 'Glycerophospholipid metabolism' is highlighted in bold in Table S1, to guarantee uniformity of content.

The authors’ answer: Thank you for your suggestion. We have now formatted Table S2 to highlight the term 'Glycerophospholipid metabolism' in bold to ensure content uniformity. (line 1201)

13.As illustrated in Table 4, it is imperative to specify the definitions of the various antibiotic susceptibilities, namely S, R and I.

The authors’ answer: Thank you for your suggestion. We have now specified the definitions of the various antibiotic susceptibilities (S, R, and I) in Table 4. (line 702)

14.Although the study is very comprehensive and provides a thorough discussion of the topic, it lacks an identification of the weaknesses and limitations of the research conducted, as well as any prospects for future research. I therefore suggest mentioning these at the end of the Results and Discussion section.

The authors’ answer: Thank you for your suggestion. We have added a discussion of the weaknesses, limitations, and future research prospects at the end of the Results and Discussion section.

Lines 460-462: Future studies will functionally validate L. plantarum B22 lipase genes through heterologous expression.

Lines 754-757: Future studies will employ heterologous expression combined with X-ray crystallography and cryo-electron microscopy (cryo-EM) to elucidate the structure and catalytic mechanism of L. plantarum B22 esterase/lipase. Gene editing of key targets will further validate strain-specific probiotic functionality and safety.

15.Conclusions

The research conclusions are a concise and clear summary of the results obtained, addressing the initial objectives and answering the formulated research questions. I therefore consider them to be very appropriate and in line with the work carried out.

The manuscript is of high quality and makes a significant contribution to the field of probiotic food safety assessment by using genomic tools. However, issues of methodological transparency must be addressed thoroughly. Attention must also be paid to the statistical procedures applied, the labelling of figures and how they are presented

Taking the above into account, I believe that the manuscript requires only minor revisions.

I look forward to hearing from you in response to my concerns.

The authors’ answer: Dear reviewer, thank you very much for your positive evaluation of our manuscript. We will thoroughly address these issues in the revisions. Specifically, we will:

  • Provide more detailed descriptions and references of our methodology to ensure complete transparency.
  • Improve the labelling of figures and their presentation to make them more informative and easier to understand.

Thank you again for your valuable feedback. We will respond promptly with our revisions.
